# A magma ocean origin to divergent redox evolutions of rocky planetary bodies and early atmospheres

Jie Deng [1,4✉], Zhixue Du [2✉], Bijaya B. Karki[3], Dipta B. Ghosh[3] & Kanani K. M. Lee [1,5]

Magma oceans were once ubiquitous in the early solar system, setting up the initial conditions for different evolutionary paths of planetary bodies. In particular, the redox conditions of magma oceans may have profound influence on the redox state of subsequently formed mantles and the overlying atmospheres. The relevant redox buffering reactions, however, remain poorly constrained. Using first-principles simulations combined with thermodynamic modeling, we show that magma oceans of Earth, Mars, and the Moon are likely characterized with a vertical gradient in oxygen fugacity with deeper magma oceans invoking more oxidizing surface conditions. This redox zonation may be the major cause for the Earth's upper mantle being more oxidized than Mars' and the Moon's. These contrasting redox profiles also suggest that Earth's early atmosphere was dominated by $CO_2$ and $H_2O$, in contrast to those enriched in $H_2O$ and $H_2$ for Mars, and $H_2$ and $CO$ for the Moon.

[1] Department of Geology and Geophysics, Yale University, New Haven, CT 06511, USA. [2] State key Laboratory of Isotope Geochemistry, Guangzhou Institute of Geochemistry, Chinese Academy of Sciences, 510640 Guangzhou, China. [3] School of Electrical Engineering and Computer Science, Department of Geology and Geophysics, and Center for Computation and Technology, Louisiana State University, Baton Rouge, LA 70803, USA. [4] Present address: Earth, Planetary, and Space Sciences, University of California, Los Angeles, CA 90095, USA. [5] Present address: Lawrence Livermore National Laboratory, Livermore, CA 94550, USA. ✉email: jie.deng@aya.yale.edu; duzhixue@gig.ac.cn

The redox condition of planetary bodies influences their chemical differentiation and governs the composition of overlying atmospheres[1–5]. For instance, to understand how bio-essential volatiles such as carbon and hydrogen were initially incorporated near Earth's surface requires knowledge about the redox state during early stages of Earth's history. A number of studies have shown that the uppermost mantle of present-day Earth is considerably oxidized (IW + 3.5, that is, 3.5 log units above the iron-wüstite buffer)[6]. Petrological evidence also suggests that such oxidized conditions formed early, 4.3–4.4 Ga ago[7]. Unlike Earth, the present-day Martian and Lunar mantles are considered to be much more reduced (~IW − 1)[8–11].

These contrasting oxidization states may have been set up during the early phase of planetary formation when magma oceans (MOs) could have existed[3,12]. Such an MO involved mechanism has yet to be fully established because relevant redox controlling reactions are still poorly constrained in realistic magma ocean scenarios. Several studies have inferred the oxygen fugacity ($f_{O_2}$) profile of silicate melts at high pressures using the experimental data at zero or relatively low pressures and is applicable only for shallow magma oceans[13,14]. The oxygen fugacity is a function of pressure, temperature, and composition, thus likely varying greatly within MOs that could have extended very deep, even covering the entire mantle regime.

Here, we study the redox controlling reactions in magma oceans by simulating silicate melts containing ferrous and ferric iron with first-principles molecular dynamics (FPMD) and perform thermodynamic modeling at pressures that cover the entire Earth's mantle and temperatures up to 5000 K. The results suggest that ferric iron becomes increasingly energetically favorable with pressure mainly due to its small partial molar volume in silicate melts under large compression. Consequently, the magma oceans of Earth, Mars, and the Moon, if compositionally homogeneous due to vigorous mixing, would be characterized with a vertical gradient in oxygen fugacity. Specifically, a deeper magma ocean existing in the early Earth would have more oxidizing surface conditions compared with those of smaller bodies like Mars and the Moon. The contrasting surface conditions between these planetary bodies suggest that the early atmosphere in equilibrium with Earth's surface may have been dominated by $CO_2$ and $H_2O$, in contrast to those enriched in $H_2O$ and $H_2$ for Mars and $H_2$ and CO for the Moon.

## Results and discussion

**Equations of state of silicate melts.** At the base of a MO where metallic melts may pond before sinking into the core[15], the oxygen fugacity is governed by the equilibrium between the metallic and silicate melts, and can be directly calculated given the compositions of these melts are known. Away from the base where metallic melts is absent due to its rapid sinking velocity[16], the MO redox state is controlled by the following redox buffering reaction[3,17]:

$$FeO(melt) + \frac{1}{4}O_2 = FeO_{1.5}(melt) \tag{1}$$

The thermodynamic behavior of the above reaction informs how oxygen fugacity varies with temperature and pressure. Taking the oxygen fugacity at the MO base as the boundary condition, one may, in principle, obtain the oxygen fugacity throughout the MO if the thermodynamic properties of the reactants and products in Eq. 1 are known. One key-parameter is the difference in molar volumes between $FeO_{1.5}$ and FeO in the melts, $\Delta V$. Its value has been directly measured only at 1 bar[18] and also inferred from experiments performed up to 23 GPa and ~2500 K[12,13,18–23]. However, these conditions are still far from what are expected in MOs of Earth and Mars. Our goal is to

calculate $\Delta V$ as a function of pressure, temperature, and composition so that we can constrain oxygen fugacity in the redox buffering Eq. (1) under directly applicable conditions. Moreover, we evaluate the MO redox states of Earth, Mars, and the Moon in order to understand their oxidation conditions of the present-day mantle and the chemistry of earliest atmosphere.

We first present the results from FPMD simulations of iron-bearing $MgSiO_3$ liquids with iron in different valence states at 2000–4000 K and up to 140 GPa (Methods). The calculated pressure–volume–temperature ($P$–$V$–$T$) relationships can be described with the following equation:

$$P(V, T) = P(V, T_0) + B_{TH}(T - T_0) \tag{2}$$

Here $P(V, T_0)$ represents the reference isotherm at $T_0 = 3000$ K using a fourth-order Birch-Murnaghan equation of state. The second term contains a thermal pressure coefficient, $B_{TH}(V) = \left[a - b\left(\frac{V}{V_0}\right) + c\left(\frac{V}{V_0}\right)^2\right]/1000$ where $a$, $b$, and $c$ are constants for a given melt composition. The bulk moduli of the $Fe^{2+}$-bearing melts are systematically larger than those of the $Fe^{3+}$ bearing melts (Supplementary Table 1). This means that the $Fe^{3+}$ bearing melts are more compressible at the conditions investigated (Fig. 1), consistent with previous low-pressure studies[18,24].

Using the pressure–volume results of simulated silicate melts for the same molar content of $Fe^{3+}$ and $Fe^{2+}$, we calculate the difference in molar volume ($\Delta V$) between $FeO_{1.5}$ and FeO in the melts as a function of pressure (Fig. 1). Our calculated value of $\Delta V$ at zero pressure agrees well with existing experimental data[18,24] (Supplementary Fig. 1). As pressure increases, $\Delta V$ decreases rapidly initially in the low-pressure regime. Thereafter, $\Delta V$ increases slightly and then decreases gradually at higher pressures. The predicted non-monotonic pressure trend weakens at higher temperatures. For silicate melts of different iron contents (i.e., 12.5 and 25 mol%), $\Delta V$ takes slightly different values, showing a weak positive trend with iron content. This is consistent with the observed weak dependency of $\Delta V$ on the melt composition[18]. Our results thus show that $\Delta V$ remains positive at all pressures up to 140 GPa irrespective of temperature and composition. This finding contradicts previous inferences that $\Delta V$ would keep on decreasing and eventually become negative within the pressure range of Earth's mantle[3,12].

Previous models on $\Delta V$ either adopt a bulk modulus derivative of 4 or use an equation of state fit to experimental data within a limited pressure range[12–14,23]. We compare model values with our calculated results for silicate melts of 12.5 mol% iron, as these models are designed for Earth's relevant composition (Supplementary Fig. 1). At low pressures (<10 GPa), our results are in good agreement with the recent model by ref. [12], both showing a sharp decrease of $\Delta V$ at low pressures, whereas at higher pressures, our results are in better line with other earlier models[13,14,23], all showing that $\Delta V$ gradually levels out. These $\Delta V$ differences arise mainly due to the different pressure dependencies of the incompressibility ($K'$) of $FeO_{1.5}$ and FeO in silicate melts adopted by the previous studies. Our 4th order Birch-Murnaghan fit yields a lightly larger $K'$ for $FeO_{1.5}$ (4.6) than that of FeO (3.3). Previous studies other than ref. [12] assume $K'$ of $FeO_{1.5}$ and FeO to be 4, thus exhibiting similar pressure dependency of $\Delta V$ to our study. The contrasting behavior of $\Delta V$ from ref. [12] is caused by drastically different $K'$ values, 1.3 and 8, respectively, for $FeO_{1.5}$ and FeO. These extreme values of $K'$ are not consistent with other experimental studies on silicate melts for which $K'$ is 3–8[25,26] and on FeO liquid for which $K'$ is 3–4[27,28]. The reason for this inconsistency is, however, unclear. Our analysis of the coordination environment of iron in the

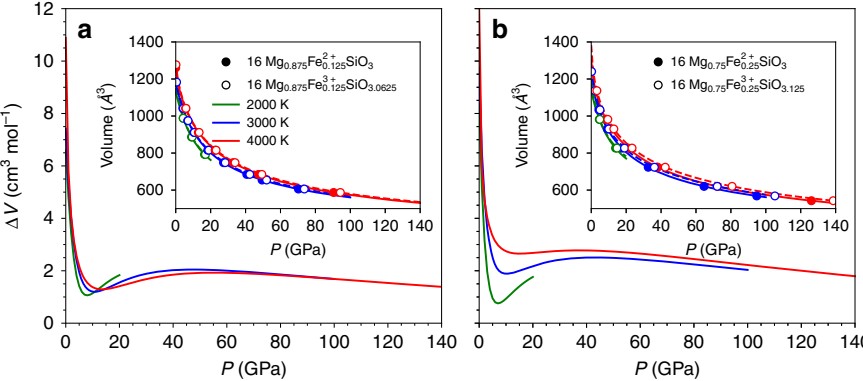

**Fig. 1 Molar volume difference (ΔV) between FeO_{1.5} and FeO in silicate melts.** The calculated ΔV is shown as a function of pressure at different temperatures: **a** 12.5 mol% iron for Earth- and Moon-like magma ocean and **b** 25 mol% iron for Mars-like magma ocean. Insets show the corresponding pressure–volume relationships for melts containing 12.5 and 25 mol% iron as $Fe^{2+}$ (solid symbols and curves) and $Fe^{3+}$ (open symbols and dashed curves). Volumes are plotted along isotherms only to pressures where the simulated systems were in a liquid state. The 1σ standard deviation of ΔV is ~0.2–0.5 $cm^3$ $mol^{-1}$.

silicate melts shows that the mean Fe–O coordination increases rapidly initially with pressure and more gradually at pressures beyond 40 GPa (Supplementary Fig. 2). This pressure trend is similar to that of ΔV. This implies an inherent correlation between the local iron-oxygen bonding environment in the silicate melt and ΔV.

We stress that our first-principles results make no assumption on the value of K' of FeO_{1.5} and FeO, so they are directly applicable over the entire mantle regime of Earth. To explore this implication further, we evaluate ΔV along two representative magma ocean thermal profiles referred to as "cold" and "hot" hereafter (Supplementary Fig. 3). The calculated ΔV varies considerably but remains positive over wide ranges of pressure and temperature of magma ocean relevance (Supplementary Fig. 4), thus indicating a positive contribution of pressure to $f_{O_2}$.

**Redox profiles**. Our calculated ΔV profiles along the magma ocean thermal profiles are used to assess the redox state of magma oceans of relevance to Earth, Mars, and the Moon. We assume that the MOs are fully convective and well-mixed, resulting in a homogeneous $Fe^{3+}$ to the total Fe ratio ($Fe^{3+}$/ΣFe)[3]. The thermodynamic relationship for the reaction (1) is

$$-\frac{\Delta G_r^0(P_0, T) + \int_{P_0}^{P} \Delta V(P, T)dP}{RT} = \ln\frac{X_{FeO_{1.5}}^{melt}}{X_{FeO}^{melt}} + \ln\frac{\gamma_{FeO_{1.5}}^{melt}}{\gamma_{FeO}^{melt}} - \frac{1}{4}\ln f_{O_2},$$

(3)

where $\Delta G_r^0(P_0, T)$ is the free energy of the reaction (Eq.(1)) at reference pressure $P_0$(1 bar) and temperature T, X and γ are the molar fractions and activity coefficients of the Fe-oxide component, respectively, $f_{O_2}$ is the oxygen fugacity, and R is the gas constant. The above equation has been widely used in many literatures[3,12,13,23] and it suggests that the variation of $f_{O_2}$ with pressure explicitly hinges on ΔV only. However, one should note that ΔV(P, T) not only depends on pressure and temperature but also implicitly on many extensive properties, including the configuration entropy, and excess enthalpy. We first evaluate $\Delta G_r^0(P_0, T)$ for FeO_{1.5}, FeO, and O_2 as a function of temperature (Supplementary Fig. 5 and Supplementary Note 1). We then estimate the activity ratio $\ln\frac{\gamma_{FeO_{1.5}}^{melt}}{\gamma_{FeO}^{melt}}$ by relating it to the interaction parameters between all the components following ref. [19]. Moreover, the experimental results on ferric iron content ($Fe^{3+}$/ΣFe) at

various conditions (listed in Supplementary Table 2) are fit to the Eq. (3) to resolve the interaction parameters (Supplementary Table 3, Supplementary Fig. 6, Supplementary Note 2). We explore four different methods to fit the interaction parameters, but all models yield very similar redox profiles for MOs (Supplementary Fig. 7). We choose the one with smallest reduced chi-square as the best model and our predicted ferric iron contents (shown in Supplementary Fig. 8) are broadly consistent with the observations by both 1-bar experiments[18–22] and the recent high-pressure experiments[12,13,23] (Supplementary Note 3).

The redox gradients in MOs of Earth, Mars, and the Moon are calculated using Eq. (3) along a cold thermal profile where 2100 K is assumed to be temperature at the surface (Fig. 2). Similar results are obtained for a hot geotherm with the surface temperature set at 2500 K (Supplementary Fig. 9).The uncertainties of all the parameters in Eq. (3) are propagated to calculate the oxygen fugacity using *LMFIT* package[29]. We use ΔV of 12.5 mol% Fe in silicate melts as a representative value for Earth[30] and the Moon[31] and that of 25 mol% Fe for Mars[32]. This assumption is justified for given mantle compositions of these three planetary bodies (Supplementary Table 4) because of relatively small effects of iron content on ΔV (Fig. 1). We quantify the redox states in terms of oxygen fugacity relative to IW buffers, that is, ΔIW = log$f_{O_2}$ − IW, where the reference IW is taken from ref. [33]. Since the temperatures considered are higher than the temperature at which this IW buffer is calibrated, we extrapolate this buffer equation to high temperatures[12,13]. We also assume that the bases of MOs are at depths of 55 GPa[34], 14 GPa[35], and 5 GPa[36], and the corresponding redox states (ΔIW) are −2, −1.5, and −2 for Earth, Mars, and the Moon, respectively[13]. These redox values are representatives for terrestrial bodies when the molten iron ponds are assumed to be in local equilibrium with the overlaying MOs[13]. The pressures considered here are based on the single stage model and the complete equilibrium between the silicate melt and iron melt. More general consideration of magma ocean depths is discussed below.

Along both thermal profiles considered, the absolute oxygen fugacities of the MOs of Earth, Mars, and the Moon all increase with depth, though more gradually at greater depths (Supplementary Fig. 9). This is expected because ΔV decreases with increasing pressure and always remains positive over the conditions we investigate. However, the relative oxygen fugacity (ΔIW) first increases slightly with pressure by ~0.3 log unit in the uppermost mantle and then gradually decreases with pressure throughout the rest of mantle (Fig. 2). Our results show that the

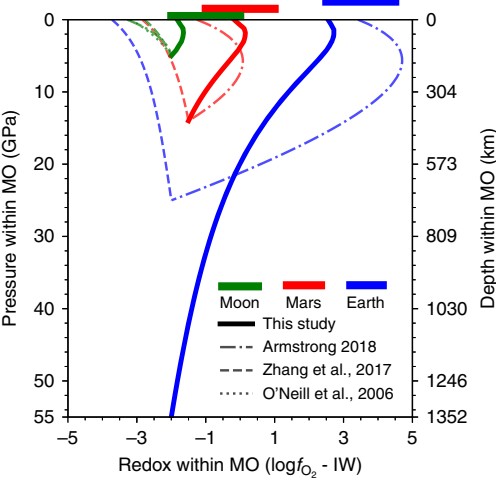

**Fig. 2 Redox profiles of magma oceans (MOs) for Earth, Mars, and the Moon.** Redox state defined by $\Delta IW = \log f_{O_2} - IW$ is shown as a function of pressure along a cold thermal profile. The MO bases are taken to be at depths corresponding to 55, 14, and 5 GPa with redox states ($\Delta IW$) of $-2$, $-1.5$, and $-2$ for Earth, Mars, and the Moon, respectively. The 1σ standard deviation of the oxygen fugacity is ~0.5 log unit for this study (thick solid curves). The previous model results from ref. [12] (dashed-dotted curves) and ref. [23] (dotted curves) are also shown within their applicable ranges. For the model of ref. [13] (dashed curves), we follow the model to extrapolate to 15 and 25 GPa to predict the oxygen fugacities of the Martian and Earth's MOs, respectively. The redox states of the present (upper) mantle of Earth, Mars, and the Moon are displayed in horizontal bars[55]. See Supplementary Fig. 9 for MO redox profiles along a hot thermal profile and Supplementary Fig. 3 for the thermal profiles.

upper mantle should have been relatively more oxidized. Therefore, an oxidized upper mantle is a natural consequence of a MO because the pressure- and temperature-dependent $\Delta V$ and $\Delta G_r^0(P_0, T)$ of the Eq. (3) make ferric iron increasingly stable at greater depths even at relatively reduced conditions (this raises $Fe^{3+}/\Sigma Fe$ of silicate melts in equilibrium with metal alloy). Additionally, our derived redox profiles of Earth, Mars, and the Moon are nearly parallel owing to the similar $\Delta V$ values. They show that Earth is ~2 log units more oxidized than Mars which, in turn, is ~2 log units more oxidized than the Moon at the same depth. This order of relative redox states of the MOs of the early Earth, Mars, and the Moon coincides with that of their present-day mantles, implying a possible inheritance of present-day oxidation states of these planets from their early MOs.

The comparison between the predicted redox profile of the MO with that of the present-day mantle for each planet informs us how the MO stage influences the subsequent redox evolution of each planet throughout its history. The oxidation state of the uppermost mantle of the present-day Earth is near $IW + 3.5$[37] and has remained constant within ~1.0 log unit since at least the early Archean[6,7]. Our predicted redox state of the uppermost MO of Earth is at the lower bound of present-day values. Likewise, the ferric iron content corresponding to this redox profile is 1.0–3.5%, overlapping with the lower end of the present-day ferric iron abundances of the upper mantle[37]. The predicted redox state and ferric iron content suggest that Earth's oxidizing uppermost mantle is a natural outcome of the thermodynamic equilibrium across the deep MO during the MO stage. Secondary contributions may arise from other mechanisms, including disproportionation of $Fe^{2+}$ in the lower mantle by crystallization of bridgmanite[38,39], and/or late accretion of oxidized materials[40,41]. Compared to the silicate Earth, the Martian uppermost mantle

is less oxidized with $f_{O_2}$ ~IW[11,42], which is consistent with our predicted redox state of the shallow Martian MO. This similarity may suggest negligible effects of subsequent tectonic processes and other oxidizing mechanisms mentioned above on the redox state of the Martian mantle[43]. Lunar basalts record oxygen fugacity ranging from IW to IW-2[8–10] and our predicted redox state falls into the lower end of the observed values. Our predicted redox profiles differ considerably from those based on previous models (Fig. 2). Previous models have generally predicted relatively more reduced MOs of the Moon and Mars and either very reducing[13] or very oxidizing MO of Earth[12]. It is important to note that the previously used data are limited with respect to pressure and temperature, for example, up to 3 GPa and 1673 K[23], 7 GPa and 2023 K[13], and 23 GPa and 2300 K[12] (Supplementary Table 3).

We also investigate the effects of varying depth of the MOs on the redox states of the surface and equilibrium ferric iron content (Fig. 3 and Supplementary Fig. 10). The redox states of the MO bases ($\Delta IW$) are assumed to be fixed at $-2$, $-1.5$, $-2$, respectively, for Earth, Mars, and the Moon. A deeper MO generally shifts upwards its oxygen fugacity profile at shallower depths (Fig. 3a). The redox states and ferric iron contents of the Lunar and Martian MOs are marginally affected due to their small sizes. In contrast, the Earth's MO may have reached 25–90 GPa based on moderately siderophile elements abundances, assuming models for single or multi-stage core formation with partial or complete equilibrium between impactor and proto Earth[34,44]. The oxygen fugacity of Earth's surface would decrease by ~1.5 log units if the base of MO moved upwards from 55 GPa to 25 GPa. Concurrently, $Fe^{3+}/\Sigma Fe$ would also drop by a factor of two. An even deeper magma ocean may induce the spin transition of iron in the silicate melts. However, the effect of the spin transition on the oxygen fugacity is shown to be insignificant within the MO thermal profiles considered here (Methods and Supplementary Fig. 11). Note our assumption that the ferric iron distribution is homogeneous within the MO due to vigorous convection. However, this ferric iron content profile likely evolves during the solidification of the MO. The evolution is controlled by how the MO crystallizes and the partitioning of iron species between the melt and crystal, which are still poorly constrained. Nevertheless, our study suggests that the whole mantles of Earth and Mars could have been as enriched in ferric iron as the present-day upper mantle since the MO stage.

**Chemistry of early atmospheres.** The redox states of the MOs may have dictated the chemical speciation of the early atmospheres. For simplicity, we consider a case where the early atmosphere is at chemical equilibrium with the underlying MO[3] and use the approach of ref. [45] to calculate the speciation of volatiles. Based on our results shown in Fig. 3, the redox state at the MO surface corresponds to $\sim IW + 2$ for Earth, $\sim IW - 0.3$ for Mars, and $\sim IW - 2$ for the Moon. Assuming a simple C–O–H atmosphere with a mass H/C ratio of 0.5 at 1 bar and 1800 K, we show that the Earth's early atmosphere would be enriched in $H_2O$ (~70 mol%) and $CO_2$ (~15 mol%) but depleted in CO and $H_2$. The early Martian atmosphere would consist of $H_2O$ and $H_2$ in almost equal amounts (each ~40 mol%), ~15 mol% CO, and ~5 mol% $CO_2$. In contrast, the early lunar atmosphere would be enriched in $H_2$ (>70 mol%) and CO (~20 mol%) and relatively depleted in $H_2O$ (10 mol%)[3,45] (Fig. 4). These early atmospheres further evolve as the planets cool down. The speciation and mass of the atmosphere would likely change over time due to the thermodynamic re-equilibrium, hydrodynamic loss, as well as subsequent degassing and ingassing/regassing. Nevertheless, these distinct early atmospheric compositions may have fundamentally

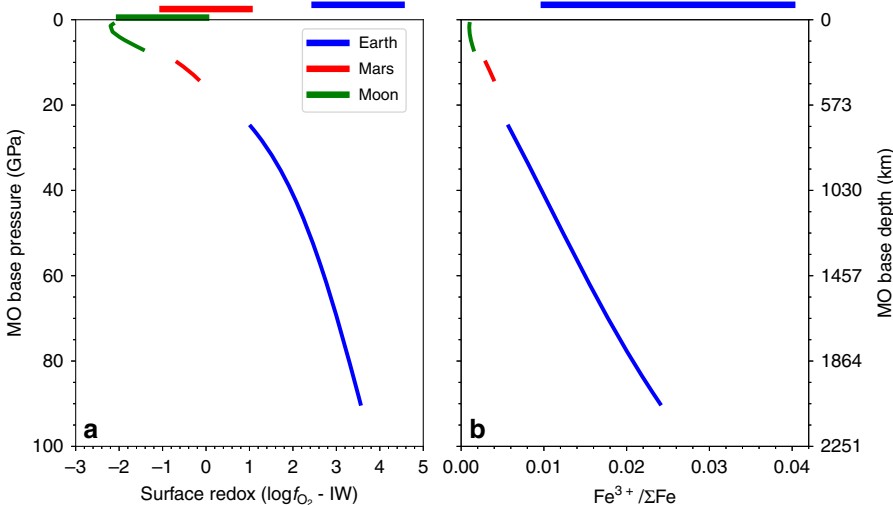

**Fig. 3 Redox state and ferric iron contents at the surfaces of magma oceans (MOs) of varying depths.** The calculated relative redox state **a** and $Fe^{3+}/\Sigma Fe$ ratio **b** of MOs of Earth (blue), Mars (red) and the Moon (green) as a function of pressure at the base of MOs considering a cold thermal profile (Supplementary Note 2). Calculations are performed at the plausible pressure ranges of the bases of the MOs suggested by previous studies[34-36,44]. The redox state/ferric iron content of the present (upper) mantle of Earth (blue), Mars (red) and the Moon (green) are presented as horizontal bars[55]. The ferric iron contents of the mantle of Mars and the Moon are poorly constrained (not shown) because the available samples suffer from alterations and post-formation oxidations and cannot reflect the ferric iron contents of the source mantle[56,57]. The 1σ standard deviation is ~0.5 log unit for of the oxygen fugacity and ~0.03–0.06 for the ferric iron content.

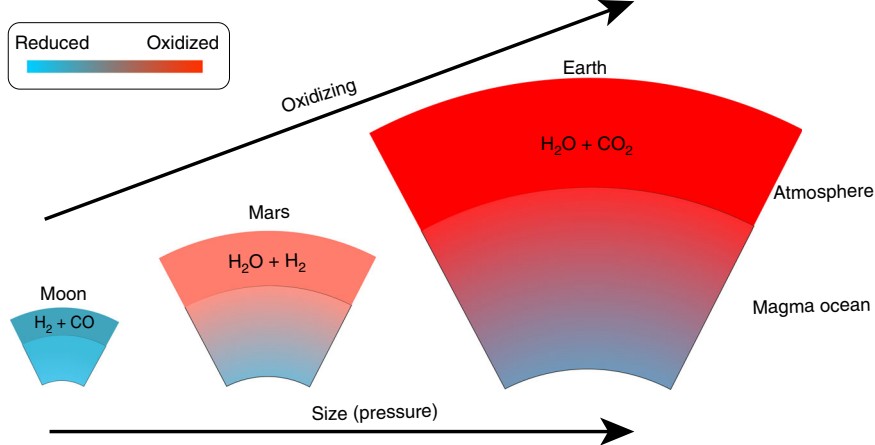

**Fig. 4 Inferred compositions of early atmospheres.** The redox states of magma oceans and the dominant chemical speciation of the overlying atmospheres are shown for Earth, Mars and the Moon. Refer to the text for estimated fraction of each species. The thicknesses and oxidation states of atmospheres are not scaled.

influenced the subsequent evolution of these terrestrial planets, including climate, magma ocean solidification, and the evolution of surficial conditions[3,46].

The vertical gradient in the MO redox state predicted here may also apply to other rocky planets where MOs were once formed. For example, Earth's sister planet Venus is of similar size and has similar iron content. The redox state of the post-MO upper mantle of Venus, to first order, may be similar to that of Earth and could be tested by future Venusian missions. In addition, super-Earths close to their host stars may have MOs extended to various depths and their atmospheres can potentially be detected in the near future with space telescope missions[47].

## Methods

**Computational details.** First-principles molecular dynamics (FPMD) simulations were carried out using the VASP software[48] in the *NVT*-canonical ensemble with temperature controlled by a Nosé thermostat[49]. The projector augmented wave potentials[50,51] were employed together with the generalized gradient approximation (GGA) to the exchange-correlation potential[52]. The plane-wave basis cutoff was set at 400 eV and Brillouin zone sampling was performed at the Gamma point. Pulay stress corrections were applied to the calculated pressures. We simulated $Mg_{14}Fe_2Si_{16}O_{48}$(ferrous) and $Mg_{14}Fe_2Si_{16}O_{49}$(ferric) melts for 12.5 mol% iron and $Mg_{12}Fe_4Si_{16}O_{48}$(ferrous) and $Mg_{12}Fe_4Si_{16}O_{50}$(ferric) melts for 25 mol% iron. Iron was set to be in high-spin state in all simulations. At each volume, the system was initially melted and thermalized at 6000 K, and then subsequently quenched down to desired lower temperatures of 4000, 3000, 2500, and 2000 K. Simulations were run for 10–30 picoseconds with time step of 1 femtosecond. Further details can be found in ref. [53].

**Calculation of volume difference (ΔV).** The molar volume difference between $FeO_{1.5}$ and FeO defined as $\Delta V = V_{FeO_{1.5}} - V_{FeO}$ is calculated as the volume difference between the ferric and ferrous iron-bearing silicate melts. Take the silicate melt with 12.5 mol% iron as an example. The volumes of $Mg_{14}Fe_2Si_{16}O_{48}$ and $Mg_{14}Fe_2Si_{16}O_{49}$ melts referred to as $V_{Mg_{14}Fe_2Si_{16}O_{48}}$ and $V_{Mg_{14}Fe_2Si_{16}O_{49}}$, respectively, are calculated at the same pressure and temperature conditions using the resolved equation of state parameters (Supplementary Table 1). These volumes can be

related to the partial volumes of components by:

$$V_{Mg_{14}Fe_2Si_{16}O_{48}} = 14V_{MgSiO_3} + 2V_{FeO} + 2V_{SiO_2} + V^{E,reduced} \tag{4}$$

and

$$V_{Mg_{14}Fe_2Si_{16}O_{49}} = 14V_{MgSiO_3} + 2V_{FeO_{1.5}} + 2V_{SiO_2} + V^{E,oxidized}, \tag{5}$$

where $V^{E,reduced}$ and $V^{E,oxidized}$ are the excess volumes for reduced and oxidized systems, respectively, and are sensitive to the amount of iron. Several previous low-pressure experiments show that the excess terms are small for silicate melts if $Na_2O$, $Al_2O_3$, and $TiO_2$ components are absent[18,24]. In this case, we can approximate $\Delta V$ by

$$\Delta V = V_{FeO_{1.5}} - V_{FeO} \approx \left(V_{Mg_{14}Fe_2Si_{16}O_{49}} - V_{Mg_{14}Fe_2Si_{16}O_{48}}\right)/2 \tag{6}$$

Similarly, for 25 mol% iron content, we use

$$\Delta V = V_{FeO_{1.5}} - V_{FeO} \approx \left(V_{Mg_{12}Fe_4Si_{16}O_{50}} - V_{Mg_{12}Fe_4Si_{16}O_{48}}\right)/4. \tag{7}$$

By using the above equation to calculate $\Delta V$, we assume that $V^{E,oxidized}$ and $V^{E,reduced}$ take small similar values so $V^{E,oxidized} - V^{E,reduced} \approx 0$. If $V^{E,oxidized} - V^{E,reduced}$ is a large non-zero value, one would expect that the $\Delta V$ differs significantly between the two compositions considered (12.5 and 25 mol% iron in silicate melts). However, our calculated results show that the $\Delta V$ values of 12.5 and 25 mol% iron contents differ slightly from each other and the difference diminishes especially at high pressures, which justifies our assumptions.

It should be noted that the small excess volume is not conflicted with the large Margules interaction parameters resolved for silicate melts. The excess volume is thermodynamically defined as $V^E = \left(\frac{\partial G_{mix}}{\partial P}\right)_T = \left(\frac{\partial H_{mix}}{\partial P}\right)_T$, where $G_{mix}$ and $H_{mix}$ are the Gibbs free energy and enthalpy of mixing, respectively; $P$ is pressure; and $T$ is temperature. $H_{mix}$ is a function of interaction parameter ($W$) and composition[54]. For a binary system with endmember components A and B, $H_{mix} = WX_AX_B$, where $X_A$ and $X_B$ are the molar fractions of A and B, respectively. Therefore, a small $V^E$ requires that the pressure derivative of the interaction parameter to be small but does not necessarily mean that the value of $W$ is small. Indeed, both in this study and many other studies[13,23], $W$ is assumed to be pressure independent, which is in line with the assumption that $V^E$ is small.

**Effects of spin transition of iron on $\Delta V$ and oxygen fugacity**. Both ferric and ferrous irons undergo electronic spin transitions at high pressure as predicted by a recent FPMD study[53]. The high- to low-spin transition of $Fe^{3+}$ and $Fe^{2+}$ occurs gradually over pressure intervals centered around 90 and 110 GPa, respectively, at 3000 K. These transition pressures are higher than the maximum pressures of the magma oceans considered in this study (Fig. 2). As both $Fe^{3+}$ and $Fe^{2+}$ will be mostly in high-spin (HS) state at relevant magma ocean pressures, we evaluate the volume difference between $FeO_{1.5}$ and $FeO$ as $\Delta V = V_{HS}^{FeO_{1.5}} - V_{HS}^{FeO}$. However, all $Fe^{3+}$ and $Fe^{2+}$ will not undergo the HS-LS transition at a given condition. This means that the spin transition-induced changes in volume also contribute to our $\Delta V$ evaluation. We assess the spin effects on $\Delta V$ using the spin phase diagrams from Karki et al.[53]. Considering exact HS and LS distributions for both ferrous and ferric irons, we can evaluate the volume difference between $FeO_{1.5}$ and $FeO$ as

$$\begin{aligned}\Delta V_{exact} = &\left(V_{HS}^{FeO_{1.5}} - V_{HS}^{FeO}\right) - n_{LS}^{Fe^{3+}}\left(V_{HS}^{FeO_{1.5}} - V_{LS}^{FeO_{1.5}}\right) \\ &+ n_{LS}^{Fe^{2+}}\left(V_{HS}^{FeO} - V_{LS}^{FeO}\right),\end{aligned} \tag{8}$$

where $V_{HS}^{FeO_{1.5}} - V_{HS}^{FeO} = \Delta V$ has been rigorously constrained in this study. $n_{LS}^{Fe^{3+}}$ and $n_{LS}^{Fe^{2+}}$ represent the fractions of $Fe^{3+}$ and $Fe^{2+}$ in low-spin (LS) state, respectively (satisfying the relations $n_{HS}^{Fe^{3+}} + n_{LS}^{Fe^{3+}} = n_{HS}^{Fe^{2+}} + n_{LS}^{Fe^{2+}} = 1$, where $n_{HS}^{Fe^{3+}}$ and $n_{HS}^{Fe^{2+}}$ represent the corresponding HS fractions) and their values as a function of pressure and temperature for silicate melt with 25% iron can be found in ref. [53]. Karki et al.[53] also evaluated the $V_{HS}^{FeO_{1.5}} - V_{LS}^{FeO_{1.5}}$ and $V_{HS}^{FeO} - V_{LS}^{FeO}$ to be constant with respect to pressure within the computational uncertainties. At 3000 and 4000 K, $V_{HS}^{FeO_{1.5}} - V_{LS}^{FeO_{1.5}} \approx 1.25$ cm$^3$ mol$^{-1}$ and 1.00 cm$^3$ mol$^{-1}$, respectively, and $V_{HS}^{FeO} - V_{LS}^{FeO} \approx 1.75$ cm$^3$ mol$^{-1}$ and 1.10 cm$^3$ mol$^{-1}$, respectively[53]. We calculate the difference of $\Delta V_{exact}$ and $\Delta V$ at 3000 and 4000 K as well as the difference of the oxygen fugacity using these two volume differences (Supplementary Fig. 11).

At pressures less than 60 GPa, we find that the deviation of the volume difference caused by considering the spin transition is less than 3%, so the oxygen fugacity does not change much when spin effects are included (Supplementary Fig. 11). With increasing pressure, the magnitude of ($\Delta V_{exact} - \Delta V$) further increases and bounces back at around 100 GPa, at which the fraction of low-spin $Fe^{3+}$ reaches around 50%. Note that at pressures greater than 80 GPa, the temperature of the MO is around 3500 K for a cold thermal profile and continues increasing with pressure. Therefore, the results at 4000 K are more relevant at these pressures. Overall, neglecting the spin transition of Fe tends to overestimate the oxygen fugacity, especially at high pressures. The maximum deviation occurs around 120 GPa, which is ~0.6 log units, comparable to the uncertainties of our model prediction (~0.5). Therefore, we consider the effects of spin transition of iron on the redox state of MOs to be mostly insignificant.

## Data availability

Authors can confirm that all relevant data are included in the paper and/or its supplementary information files.

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

## Acknowledgements

The research is supported by NSF grants (EAR-1321956 and EAR-1551348 to K.K.M.L. and EAR 1764140 to B.B.K.) and from Chinese Academy of Sciences (No. 29Y93301701 and 51Y8340107 to Z.D.). We benefited from discussions with Colin Jackson and Lars Stixrude. The computing resources were provided by the Yale Center for Research Computing (thanking Kaylea Nelson for guidance) and Louisiana State University High Performance Computing. K.K.M.L.'s effort was partially supported under the auspices of the U.S. Department of Energy by Lawrence Livermore National Laboratory under Contract DE-AC52-07NA27344.

## Author contributions

J.D. and Z.D. conceived and designed the project. J.D. performed the simulations and thermodynamic modelling. J.D., Z.D., B.B.K., D.B.G., and K.K.M.L. contributed to the analysis and manuscript preparation.

## Competing interests

The authors declare no competing interests.
