## [Peer Review File · Nature Communications]

Reviewers' Comments:

Reviewer #1:

Remarks to the Author:

Review of Deng et al. "A magma ocean origin to divergent redox evolutions of Earth, Mars, and Moon: Implications for early atmospheres" for Nature Communications.

Marc Hirschmann

Deng et al. present first principles (DFT) calculations to explore the gradient of redox potential in a homogeneous magma ocean owing to differences in the equations of state (EOS) of Fe²⁺ and Fe³⁺ in silicate liquid. They find that surface conditions are more oxidized for MO that equilibrate with Fe metal at higher pressures, and that this has significant influence on a number of important parameters affecting the early evolution of terrestrial planets, most particularly affecting early atmospheres.

This is a topic of great interest to those studying early planetary evolution and the submitted manuscript has a number of novel contributions. Unfortunately for the authors, the fundamental ideas and conclusions of this paper have already been published elsewhere. The basic idea of redox gradients in magma oceans and the implications for early atmospheres was proposed as a speculation by Hirschmann (2012) and affirmed by experiments published recently by Armstrong et al. (2019) in Science. Based on experiments, Armstrong et al. reach much the same conclusions as Deng et al., and discuss many of the same implications. Zhang et al. (2017) also discussed many of the same concepts, particularly for smaller planets, though the low pressure experiments in that work did not suggest more oxidized shallow MO conditions.

The remaining innovations of the submitted manuscript are:

- (1) The DFT calculations go to higher pressure than the experiments of Armstrong et al. (2019) and broadly affirm the conclusions based on experimental work at lower pressure, thereby making a stronger case that indeed, MO on larger planets produce comparatively oxidized surface conditions, even though the MO likely equilibrated with metal.
- (2) the nature of DFT calculations gives us greater insight into the microscopic mechanisms leading to the differential EOS of Fe²⁺ and Fe³⁺.
- (3) An additional important advance in the present manuscript is calibration of an EOS that may be more robust at high pressure than previous work. In particular, the authors highlight that the EOS employed by Armstrong et al. (2019) may not give accurate extrapolations to high pressure.

These are significant contributions. Together, these may overcome the hurdle of novelty for publication in a high profile journal such as Nature Communications, even though the essential premise and conclusions of the paper are not new.

I am not qualified to judge the specifics of the DFT calculations, though I have no reason to question their accuracy. Prior to publication, other referees with greater knowledge of DFT calculations (including those involving Fe, which are generally quite challenging), should be consulted.

I confess that I have not taken a great deal of time to examine the technical details of the calculations in this paper. Probably greater attention is warranted. One question that does arise is that in Fig. S5, the thermodynamic model has systematic differences with the high pressure experiments as a function of Fe³⁺/Fe^T. This is expressed as a slope different from unity for the high pressure experiments in this plot. This is a worrisome discrepancy. The authors suggest that the high pressure experiments of Armstrong et al. (2019) may have been biased by the effects of crystallization. However, my reading of the Armstrong et al. (2019) paper indicates that these effects, if any, are likely small. Without delving too deeply into this problem, I conclude that

whereas the DFT calculations and experiments are broadly consistent and lead to very similar conclusions, a quantitatively robust model must await further experiments at high pressure.

The last sentence of the main text comes perilously close to plagiarism. I have not done a systematic check to determine if there are other such instances.

Deng et al. "Nevertheless, these distinct compositions of early atmospheres may have set these terrestrial planets on different evolutionary paths, differing in their preservation potential of the atmospheres, climatic evolutions, ... and conditions conducive to prebiotic chemistry."

Hirschmann (2012) p. 52. "This may set terrestrial planets on different evolutionary paths, with differences for the preservation potential of the atmospheres, the evolution of their climate, and development of conditions conducive to prebiotic chemistry."

Reviewer #2:

Remarks to the Author:

The manuscript by Deng et al. presents new first-principles calculations of thermodynamic parameters controlling magma ocean redox state. They apply these calculations to hypothetical MOs on Earth, Mars and the Moon, and find that their measured upper mantle redox states essentially match their calculation and model predictions. This leads them to conclude that planetary redox conditions are natural outcome of MO thermodynamics.

I admit I was skeptical when I received this manuscript because the conclusions seemed similar to a number of other studies published in the last few years. So I didn't know what new ideas this work would bring to the table. However, the authors do a very good job of describing those previous works, explaining how the present work is new and different (which seems to mainly be the high pressure and temperature determination of ΔV), and why the other works get the results they do, whether similar or different to the present work. I believe this will be an important paper and recommend its publication with only some minor changes detailed below.

Jonathan Tucker

One general comment: I suggest the authors carefully read the manuscript with an editorial eye for grammar and usage. A few examples: "the Moon" (78 and many others; see NASA style guide <https://www.history.nasa.gov/styleguide.html>); "oxygen fugacity" (50); errant period (120); iron and wüstite are not proper nouns (163).

Other comments by line number:

41: Please provide references for the numerical value stated (I think this statement has a reference later on?)

73: This sentence confuses me. Are you saying that different studies predict different redox states for the top or a terrestrial (or martian) MO? If so, that should be clearly stated as the main impetus for this work.

176: I'm not sure this is "generally believed" as stated. It has been hypothesized. The argument is based on the assumption that bridgmanite crystallization causes disproportionation (which only recently has a little experimental evidence), and that the effects of that crystallization are felt elsewhere in the MO. But regardless, this is not a fair comparison to the present work because the present work does not consider the effects of a crystallizing magma ocean on fO_2 , as Wade and Wood does, only a fully molten MO.

200: A brief scan of ref 39 seems to indicate that they are describing hydrodynamic escape following H₂ outgassing...hydrodynamic escape on Earth has essentially been disproved (Marty 2012), and you are not arguing for an H₂-dominated terrestrial atmosphere. So I don't think this is the right reference. H₂ loss (though not hydrodynamic escape) following H₂O photolysis is a possible way to oxidize the surface (see Zahnle et al 2013 Chem Geol, or many other refs dating back to Holland in the 1960s) but may not have any effect on the mantle.

209: (A comment on Figure 2) A second y-axis of depth might help a broader readership (this could apply to Fig. 3 as well).

224: I'm not sure this is a fair criticism of the recent Armstrong work. Did they really predict IW+25 and 3+/T of 1? Or is this your modeling of their data based on a deeper and hotter MO than they assumed? I doubt IW+25 was really their model, so please be careful not to mischaracterize their work. This same comment applies to the caption of Figure 2 (e.g. line 213)—are the curves really their models, or your model of their data and assumptions?

230: (A comment on Figure 3) The axes in Fig. 3a are the same as Fig. 2, curves are the same colors, and overall appearance is similar, but that's misleading because Fig. 2 is a depth profile, and Fig. 3 is not. If someone just glances at Fig. 3 without reading the text carefully, they would probably think Fig. 3 are depth profiles as well, whereas they are really the surface condition of MOs with different base pressures. Two possible suggestions to allay this confusion are to plot Fig. 3 as separated points rather than connected lines, which would emphasize that each point is an individual calculation, or label the y-axes differently and more descriptively, for example, Fig 2 could be "P within the magma ocean" and Fig 3 could be "Magma ocean base P".

261: Where are you showing this? Figure 4 is just schematic. There are no calculations I can find in the paper or supplement to support the numbers quoted here.

279: What about Mercury or Vesta? Shallow MO and reduced mantle seems to fit this story (and has been noted in other similar publications like Hirschmann 2012 and Armstrong 2019).

Reviewer #3:

Remarks to the Author:

This is certainly a very interesting paper on an important subject. It is concise and makes important comparative predictions about the consequences of magma oceans on Earth, Mars and the Moon. But there are a number of problems in terms of detail and implicit assumptions which are not explained. If the paper is to be accepted far more detail needs to be provided about the methods employed and information about what the molecular dynamics simulations predict for the structure of the melts and the iron components at high pressure should be included. Without this information the paper is impossible to evaluate. Furthermore it is not clear what the authors are implying with their magma ocean scenario which appears over simplistic and has implicit assumptions which are not made clear. There are a number of other points that would have to be addressed before publication.

There are virtually no details given as to how these calculations are performed. Much more detail has to be provided in the SI. For example how are the volumes of the individual components determined? The volumes that are used in the thermodynamic calculations are for the standard state end member components in the pure state. But they determine these volumes from melts that are more complex so they must have a way of dealing with the molar volume of mixing. This may well be quite simple but it is totally unclear how it is performed.

How are the uncertainties calculated and why are they not propagated into the MO calculations. This propagation needs to be performed so the model curves in figure 2 can be correctly evaluated.

What is implied for the structure of the iron components in the silicate melt? The simulations

should provide important information on melt structures which would provide at least some information upon which to evaluate the results. Why is no information on melt structure provided? If the simulations capture correctly the thermodynamic state of the iron components in the melt why do the authors not account for the spin transitions? Why ignore this important change in electronic properties and just write it off as being small and inconsequential?- for which actually no basis is given.

The reference at line 308 is incorrect.

It is not clear what ΔV is being referred to at line 312. The ΔV of the HS-LS transition could well be of the order of 1-2 % if this is what the authors mean- but the effect on the ΔV of equation (1) would be larger because the transitions do not occur at the same conditions. If the authors want to calculate the effects on the volume components in the lower mantle realistically, then this should be done comprehensively otherwise what is the point of doing such calculations. The argument that the MO only extends to 60 GPa is based Ni-Co partitioning between mantle and core and is by far not accepted as a firm constraint on the depth of a MO. Figure S3 is totally misleading because it intentionally ignores the potential effects of spin transitions.

The authors use their calculations simply to calculate the volumes of the iron components and then use these data in combination with one bar thermodynamic data to determine the oxygen fugacity of a melt as a function of the ferric/ferrous ratio at high pressure. But why use the one bar data at all- why don't they calculate the oxygen fugacity at high pressure from their ab initio calculations directly from information on the free energy of the melt- if these calculations truly capture the thermodynamics of silicate liquids. Why use 1 bar interaction parameters, for example, when information to constrain these parameters at high pressure could be obtained through the excess volumes of mixing. If the authors cannot do this then they should state it and estimate what the uncertainty is likely to be as a result- then propagate this uncertainty in their calculation.

How can the authors be sure that their determined volume change describes and encapsulates completely the expected changes in the ferric/ferrous components in the liquid as a function of pressure? For experimental data the fitting of a volume change to parameterise the pressure behaviour of the ferric/ferrous ratios is reasonable because the data themselves actually constrain the variations in the ratio whereas the volume is just a fitting term and may not actually correspond to the standard state volume change. There could be other factors that end up being inadvertently described by the volume term such as changes in the interaction parameters with pressure. For the experiment based models this does not matter because they are fitted to the experiments and reproduce those values but in this study the authors attribute all changes with pressure to their estimated volume changes. The extent to which they can be sure that the calculated volumes encapsulate the real ferric/ferrous ratio variations needs to be discussed in detail in the SI. Can the authors be sure that changes in electronic state, entropies and excess volumes with pressure are being meaningfully accounted for just through the volume change and if not what are the uncertainties?

The authors propose depths of "the" magma oceans for which they calculate f_{O_2} gradients. For the earth this is taken as 55 GPa using single stage models for metal silicate equilibration. The use of this value so unequivocally in the manuscript is naïve as there are many alternative hypotheses as to how the Earth obtained its Ni-Co ratio on which this estimate is based. This implies also, for example, that there was a single magma ocean with a particular depth at which metal equilibrated with the "entire" magma ocean and then no further equilibration with metal occurred in lower pressure magma oceans. Recent models that have examined this, such as Deguen et al., 2014, *EPSL* 391: 274-287, show quite clearly that it is unrealistic to assume whole mantle equilibration with metal in a magma ocean. Only part of the magma ocean would reequilibrate with iron metal as it descends. A simplified scenario is probably ok as long as the simplifications and the assumptions are all clear but to maintain dogmatically that "the" MO on earth was at 55 GPa is simply counter to a lot of evidence. There is too little discussion at the moment over the actual scenario that is being implied. Assumptions such as the "entire" magma ocean equilibrating with iron at 55 GPa as the last step in significant core formation are not explicitly stated in the manuscript and need to be explained.

This raises the further question of what is the actual scenario that the authors are proposing. How is the ferric iron content of the magma ocean actually increased between the moon, mars and

Earth? How does oxygen mass balance in their calculation? Presumably the authors are implying that before the magma ocean formed the ferric iron content of the mantle was very low- although they do not specifically state this. Therefore where does the oxygen come from to create the raised ferric iron levels in the magma ocean? Buffering with iron metal has the implication that material either initially more oxidised or more reduced attains the same level of oxidation i.e. ferric iron content after equilibration with iron metal. If material that formed the earth was initially more oxidised then it is clear that reaction with iron metal would lower the ferric iron content to a specific level by reacting $\text{Fe} + \text{Fe}_2\text{O}_3$ into FeO . But how does this buffering work if the material is more reduced before it interacts with iron metal. How does the redox buffering actually operate? This is not at all clear in the manuscript.

Armstrong et al created a model that fitted their data over the pressure range of their experiments. They did not then extrapolate this model and they state quite clearly that the properties likely change at higher pressure. It is simply not scholarly to take a model that was only calibrated and derived over a particular pressure regime and extrapolate it to over double the pressure and a much higher temperature as in Figure 2. This model and certainly that of O'Neill was never intended to be employed in that way and by doing so the authors intentionally make the models look erroneous. Furthermore the uncertainties of all these extrapolations would make them meaningless so it is questionable what purpose this serves. These models were based on experimental measurements of ferric iron contents and oxygen fugacities for which there is almost certainly a range of equation of state parameters that would fit the data- but that would extrapolate very differently. The main point of the experimental based models is that they allow the data to be interpolated as a function of oxygen fugacity over the pressure range at which they were calibrated.

Line 225 "mainly due to their unwarranted extrapolations from low P/T experimental conditions". This statement is totally misleading in its use of the word "their" and could be understood in two ways. What the authors hopefully mean is that they themselves are performing an unwarranted extrapolation- which they should not do and the curve should simply be removed. One could understand from this statement that Armstrong et al themselves extrapolated the data- which is not the case. The statement is misleading and should be clarified and the authors need only state that no meaningful results can be obtained by extrapolating the model so far out of the range at which it was determined.

Line 164 ref 32 is used to calculate IW however again the authors are extrapolating this equation of state to much higher temperatures than it was intended. This should be addressed.

Line 166 the authors use previous values for the f_{O_2} relative to IW for the earth but they should be able to calculate these values using their model and state clearly what the assumptions are. These f_{O_2} s can then be made internally consistent with the compositions in table S4.

Figure S5 is not useful as it provides no information on the conditions at which the fit is good or bad other than the information "1 bar" and "high pressure". The use of the $\text{Fe}^{3+}/\text{Fe}^{2+}$ ratio also makes it impossible to usefully evaluate the accuracy of the model. In these figures all the experimental data at high pressures seem to fall on a slope that implies some systematic deviation to the model- but it is impossible to evaluate this without a comparison that provides also P, T and f_{O_2} information which must be facilitated. Rather than showing 4 fits- that all seem to be identical anyway- the authors should allow the model to be evaluated against the actual high pressure experimental measurements of Fe^{3+} over total iron. As most of the experimental data is determined at a buffered oxygen fugacity- the model should be compared against the experimental data at the particular f_{O_2} . Showing $\text{Fe}^{3+}/\text{Fe}^{2+}$ ratio is immediately difficult to assess because it varies over 3 orders of magnitude particularly due to the changes in the total iron content of the 1 bar experiments. The authors should rather evaluate the $(\text{Fe}^{3+}/\text{Total iron})$ ratio differences between the data and model either by comparing the model values with the data points themselves or by plotting the miss fit as a function of P with information on T and f_{O_2} given.

Line 499 "we use the value after correction for crystallization"- it is not clear what this means or why the authors choose to exclude some experimental data. This has to be more clearly explained otherwise it looks like they may be excluding experimental data that do not agree with their model.

Reviewers' comments:

Reviewer #1 (Remarks to the Author):

Review of Deng et al. "A magma ocean origin to divergent redox evolutions of Earth, Mars, and the Moon: Implications for early atmospheres" for Nature Communications.

Marc Hirschmann

Deng et al. present first principles (DFT) calculations to explore the gradient of redox potential in a homogeneous magma ocean owing to differences in the equations of state (EOS) of Fe^{2+} and Fe^{3+} in silicate liquid. They find that surface conditions are more oxidized for MO that equilibrate with Fe metal at higher pressures, and that this has significant influence on a number of important parameters affecting the early evolution of terrestrial planets, most particularly affecting early atmospheres.

This is a topic of great interest to those studying early planetary evolution and the submitted manuscript has a number of novel contributions. Unfortunately for the authors, the fundamental ideas and conclusions of this paper have already been published elsewhere. The basic idea of redox gradients in magma oceans and the implications for early atmospheres was proposed as a speculation by Hirschmann (2012) and affirmed by experiments published recently by Armstrong et al. (2019) in Science. Based on experiments, Armstrong et al. reach much the same conclusions as Deng et al., and discuss many of the same implications. Zhang et al. (2017) also discussed many of the same concepts, particularly for smaller planets, though the low pressure experiments in that work did not suggest more oxidized shallow MO conditions.

The remaining innovations of the submitted manuscript are:

- (1) The DFT calculations go to higher pressure than the experiments of Armstrong et al. (2019) and broadly affirm the conclusions based on experimental work at lower pressure, thereby making a stronger case that indeed, MO on larger planets produce comparatively oxidized surface conditions, even though the MO likely equilibrated with metal.
- (2) the nature of DFT calculations gives us greater insight into the microscopic mechanisms leading to the differential EOS of Fe^{2+} and Fe^{3+} .
- (3) An additional important advance in the present manuscript is calibration of an EOS that may be more robust at high pressure than previous work. In particular, the authors highlight that the EOS employed by Armstrong et al. (2019) may not give accurate extrapolations to high pressure.

These are significant contributions. Together, these may overcome the hurdle of novelty for publication in a high profile journal such as Nature Communications, even though the essential premise and conclusions of the paper are not new.

Response: Thank you for recognizing our contributions.

I am not qualified to judge the specifics of the DFT calculations, though I have no reason to question their accuracy. Prior to publication, other referees with greater knowledge of DFT calculations (including those involving Fe, which are generally quite challenging), should be consulted.

I confess that I have not taken a great deal of time to examine the technical details of the calculations in this paper. Probably greater attention is warranted. One question that does arise is that in Fig. S5, the thermodynamic model has systematic differences with the high pressure experiments as a function of $\text{Fe}^{3+}/\text{Fe}^T$. This is expressed as a slope different from unity for the high pressure experiments in this plot.

This is a worrisome discrepancy. The authors suggest that the high pressure experiments of Armstrong et al. (2019) may have been biased by the effects of crystallization. However, my reading of the Armstrong et al. (2019) paper indicates that these effects, if any, are likely small. Without delving too deeply into this problem, I conclude that whereas the DFT calculations and experiments are broadly consistent and lead to very similar conclusions, a quantitatively robust model must await further experiments at high pressure.

Response: $\text{Fe}^{3+}/\text{Fe}^{2+}$ measured at high pressures do seem to deviate slightly from the model prediction. We added a section in the Supplementary Information (**Supplementary Note 5: Model prediction vs. high pressure experiments**) to compare the model prediction against the high-pressure experimental results. We also added a figure (**Supplementary Figure 8**) to show the ferric iron content not only as a function of pressure but also temperature. From this analysis, we confirm that our predicted values are broadly consistent with the extant experimental results.

Nevertheless, we agree that the best way of improving the model and evaluating this model against the experiments is to perform experiments at relevant high pressures.

The last sentence of the main text comes perilously close to plagiarism. I have not done a systematic check to determine if there are other such instances.

Deng et al. “Nevertheless, these distinct compositions of early atmospheres may have set these terrestrial planets on different evolutionary paths, differing in their preservation potential of the atmospheres, climatic evolutions, ... and conditions conducive to prebiotic chemistry.”

Hirschmann (2012) p. 52. “This may set terrestrial planets on different evolutionary paths, with differences for the preservation potential of the atmospheres, the evolution of their climate, and development of conditions conducive to prebiotic chemistry.”

Response: We apologize for this unintentional and fortuitous similarity between this sentence in our manuscript and that in Hirschmann (2012). We have rewritten this sentence, as follows:

“Nevertheless, these distinct early atmosphere compositions may have fundamentally influenced the subsequent evolution of these terrestrial planets, including climate, magma ocean solidification, and the evolution of surficial conditions”

Reviewer #2 (Remarks to the Author):

The manuscript by Deng et al. presents new first-principles calculations of thermodynamic parameters controlling magma ocean redox state. They apply these calculations to hypothetical MOs on Earth, Mars and the Moon, and find that their measured upper mantle redox states essentially match their calculation and model predictions. This leads them to conclude that planetary redox conditions are natural outcome of MO thermodynamics.

I admit I was skeptical when I received this manuscript because the conclusions seemed similar to a number of other studies published in the last few years. So I didn't know what new ideas this work would bring to the table. However, the authors do a very good job of describing those previous works, explaining how the present work is new and different (which seems to mainly be the high pressure and temperature determination of ΔV), and why the other works get the results they do, whether similar or different to the present work. I believe this will be an important paper and recommend its publication with only some minor changes detailed below.

Jonathan Tucker

One general comment: I suggest the authors carefully read the manuscript with an editorial eye for grammar and usage. A few examples: “the Moon” (78 and many others; see NASA style guide <https://www.history.nasa.gov/styleguide.html>); “oxygen fugacity” (50); errant period (120); iron and wüstite are not proper nouns (163).

Response: Thank you for pointing these out. We have revised accordingly.

Other comments by line number:

41: Please provide references for the numerical value stated (I think this statement has a reference later on?)

Response: Yes. We have now repeated the references later for this sentence.

73: This sentence confuses me. Are you saying that different studies predict different redox states for the top or a terrestrial (or martian) MO? If so, that should be clearly stated as the main impetus for this work.

Response: Previous studies actually seldom extrapolate their models to predict the redox states of the magma ocean for Earth and Mars. The only example we are aware of is that Zhang et al., 2017 extrapolated their model to 25 GPa and predicted that Earth’s surface would be around IW-4.5. Under such an extremely reduced condition, Fe²⁺ may not be thermodynamically stable in the silicate melt. For other models (e.g., O’Neill et al., 2006; Armstrong et al., 2019), the authors did not extrapolate their studies to higher pressures to predict the terrestrial MO’s redox state. However, if extrapolating their models to 15 GPa for Mars’ MO and 55 GPa for Earth’s MO, one would find that the surface redox of the MO is either too low or too high.

As we also discussed in the reply to reviewer #3’s comments, extrapolating the experimentally determined model beyond their applicable range is highly uncertain. Therefore, we have now deleted this statement.

176: I’m not sure this is “generally believed” as stated. It has been hypothesized. The argument is based on the assumption that bridgmanite crystallization causes disproportionation (which only recently has a little experimental evidence), and that the effects of that crystallization are felt elsewhere in the MO. But regardless, this is not a fair comparison to the present work because the present work does not consider the effects of a crystallizing magma ocean on fO₂, as Wade and Wood does, only a fully molten MO.

Response: Yes, we agree. Wade and Wood (2005) and this study consider different stages of MO evolution. We have now deleted this sentence citing the results of Wade and Wood (2005).

200: A brief scan of ref 39 seems to indicate that they are describing hydrodynamic escape following H₂ outgassing. Hydrodynamic escape on Earth has essentially been disproved (Marty 2012), and you are not arguing for an H₂-dominated terrestrial atmosphere. So I don’t think this is the right reference. H₂ loss (though not hydrodynamic escape) following H₂O photolysis is a possible way to oxidize the surface (see Zahnle et al 2013 Chem Geol, or many other refs dating back to Holland in the 1960s) but may not have any effect on the mantle.

Response: Thank you for pointing out our flawed statement: hydrodynamic loss of H₂ has been disproved for large bodies like Earth. We have now removed this mechanism as the possible secondary contribution to oxidize the Earth's mantle.

209: (A comment on Figure 2) A second y-axis of depth might help a broader readership (this could apply to Fig. 3 as well).

Response: Thanks for the suggestions. We have now added a secondary y-axis of depth for Figures 2 and 3.

224: I'm not sure this is a fair criticism of the recent Armstrong work. Did they really predict IW+25 and 3+/T of 1? Or is this your modeling of their data based on a deeper and hotter MO than they assumed? I doubt IW+25 was really their model, so please be careful not to mischaracterize their work. This same comment applies to the caption of Figure 2 (e.g. line 213)—are the curves really their models, or your model of their data and assumptions?

Response: Their model did not predict IW+25 for the surface. When we extrapolated their model to higher pressures, we found such extreme value. We agree that this comparison is not well justified because their model does not apply beyond 25 GPa. We have now only used all previous models within their applicable pressure ranges. We have revised the text and Figure 2 accordingly.

230: (A comment on Figure 3) The axes in Fig. 3a are the same as Fig. 2, curves are the same colors, and overall appearance is similar, but that's misleading because Fig. 2 is a depth profile, and Fig. 3 is not. If someone just glances at Fig. 3 without reading the text carefully, they would probably think Fig. 3 are depth profiles as well, whereas they are really the surface condition of MOs with different base pressures. Two possible suggestions to allay this confusion are to plot Fig. 3 as separated points rather than connected lines, which would emphasize that each point is an individual calculation, or label the y-axes differently and more descriptively, for example, Fig 2 could be "P within the magma ocean" and Fig 3 could be "Magma ocean base P".

Response: Thank you for the suggestion. We have now changed the labels of x, y axis of Figure 2 to "Pressure within the magma ocean" and "redox within the magma ocean" and those of Figure 3 to "Magma ocean base pressure" and "surface redox".

261: Where are you showing this? Figure 4 is just schematic. There are no calculations I can find in the paper or supplement to support the numbers quoted here.

Response: The numbers of species fractions shown in the text were calculated using the algorithm developed by Kress et al (2004) and now we added a brief description in the caption of Fig. 4 to make this more clear. The detailed calculated results have already been shown by Hirschmann 2012 (see his Figure 1), therefore, we do not elaborate on the calculation details in our paper. We only show the dominant species, as this is the most robust and relevant information for broader readership.

279: What about Mercury or Vesta? Shallow MO and reduced mantle seems to fit this story (and has been noted in other similar publications like Hirschmann 2012 and Armstrong 2019).

Response: The Mercurian mantle has very low iron content (~3 wt%). It is not clear whether or not our calculated ΔV model based on the Moon, Earth, and Mars-like compositions still apply at such low iron content. In addition, likely large amount of sulfur in the Mercurian mantle further complicates the redox

buffering in Mercurian magma ocean. As a result, we avoid discussing Mercury, although its reduced mantle seems to be consistent with our model prediction as the reviewer pointed out.

For Vesta, the size of its body is very small and the magma ocean base is typically estimated to be at ~ 0.0001 GPa to 0.1 GPa (Steenstra et al., 2016). For such a shallow depth, according to our model, the surface redox would be essentially the same as the magma ocean base. However, the redox state of the magma ocean base is not well constrained to our best knowledge. Estimation ranges from IW-4 to IW-1 (Stolper, 1977; Pringle et al., 2013; Steenstra et al., 2016). We decide not to include it due to the poorly constrained base oxygen fugacity. But we agree that the relatively reduced HED samples are qualitatively consistent with our model.

Reviewer #3 (Remarks to the Author):

This is certainly a very interesting paper on an important subject. It is concise and makes important comparative predictions about the consequences of magma oceans on Earth, Mars and the Moon. But there are a number of problems in terms of detail and implicit assumptions which are not explained. If the paper is to be accepted far more detail needs to be provided about the methods employed and information about what the molecular dynamics simulations predict for the structure of the melts and the iron components at high pressure should be included. Without this information the paper is impossible to evaluate. Furthermore it is not clear what the authors are implying with their magma ocean scenario which appears over simplistic and has implicit assumptions which are not made clear. There are a number of other points that would have to be addressed before publication.

1) There are virtually no details given as to how these calculations are performed. Much more detail has to be provided in the SI. For example how are the volumes of the individual components determined? The volumes that are used in the thermodynamic calculations are for the standard state end member components in the pure state. But they determine these volumes from melts that are more complex so they must have a way of dealing with the molar volume of mixing. This may well be quite simple but it is totally unclear how it is performed.

Response: Sorry for the confusion. We have now added a section (**Calculation of volume difference**) in the Methods section to elucidate this issue, as follows:

“The molar volume difference between $\text{FeO}_{1.5}$ and FeO defined as $\Delta V = V_{\text{FeO}_{1.5}} - V_{\text{FeO}}$ is calculated as the volume difference between the ferric and ferrous iron-bearing silicate melts. Take the silicate melt with 12.5 mol% iron as an example. The volumes of $\text{Mg}_{14}\text{Fe}_2\text{Si}_{16}\text{O}_{48}$ and $\text{Mg}_{14}\text{Fe}_2\text{Si}_{16}\text{O}_{49}$ melts referred to as $V_{\text{Mg}_{14}\text{Fe}_2\text{Si}_{16}\text{O}_{48}}$ and $V_{\text{Mg}_{14}\text{Fe}_2\text{Si}_{16}\text{O}_{49}}$, respectively, are calculated at same pressure and temperature conditions using the resolved equation of state parameters (Supplementary Table 1). These volumes can be related to the partial volumes of components by:

$$\begin{aligned} V_{\text{Mg}_{14}\text{Fe}_2\text{Si}_{16}\text{O}_{48}} &= 14 V_{\text{MgSiO}_3} + 2 V_{\text{FeO}} + 2 V_{\text{SiO}_2} + V_{\text{E, reduced}}^{\text{E}} \\ V_{\text{Mg}_{14}\text{Fe}_2\text{Si}_{16}\text{O}_{49}} &= 14 V_{\text{MgSiO}_3} + 2 V_{\text{FeO}_{1.5}} + 2 V_{\text{SiO}_2} + V_{\text{E, oxidized}}^{\text{E}} \end{aligned}$$

where $V_{\text{E, reduced}}^{\text{E}}$ and $V_{\text{E, oxidized}}^{\text{E}}$ are the excess volumes for reduced and oxidized systems, respectively, and are sensitive to the amount of iron. Many previous low-pressure experiments show that the excess terms are small for silicate melts if Na_2O , Al_2O_3 , and TiO_2 components are absent^{18,24}. In this case, we can approximate ΔV by

$$\Delta V = V_{\text{FeO}_{1.5}} - V_{\text{FeO}} \approx (V_{\text{Mg}_{14}\text{Fe}_2\text{Si}_{16}\text{O}_{49}} - V_{\text{Mg}_{14}\text{Fe}_2\text{Si}_{16}\text{O}_{48}})/2$$

Similarly, for 25 mol% iron content, we use

$$\Delta V = V_{\text{FeO}_{1.5}} - V_{\text{FeO}} \approx (V_{\text{Mg}_{12}\text{Fe}_4\text{Si}_{16}\text{O}_{50}} - V_{\text{Mg}_{12}\text{Fe}_4\text{Si}_{16}\text{O}_{48}})/4$$

By using the above equation to calculate ΔV , we assume that $V^{\text{E, oxidized}}$ and $V^{\text{E, reduced}}$ take small similar values so $V^{\text{E, oxidized}} - V^{\text{E, reduced}} \approx 0$. If $V^{\text{E, oxidized}} - V^{\text{E, reduced}}$ is a large non-zero value, one would expect that the ΔV differs significantly between the two compositions considered (12.5 and 25 mol% iron in silicate melts). However, our calculated results show that the ΔV values of 12.5 and 25 mol% iron contents differ only marginally and the difference diminishes especially at high pressures, which justifies our assumptions.

It should be noted that the small excess volume is not conflicted with the large Margules interaction parameters we resolved for silicate melts. The excess volume is thermodynamically defined as, $V^{\text{E}} = \left(\frac{\partial G_{\text{mix}}}{\partial P}\right)_T = \left(\frac{\partial H_{\text{mix}}}{\partial P}\right)_T$, where G_{mix} and H_{mix} are the Gibbs free energy and enthalpy of mixing respectively; P is pressure; and T is temperature. H_{mix} is a function of interaction parameter (W) and composition⁵⁷. For a binary system with endmember components A and B, $H_{\text{mix}} = WX_{\text{A}}X_{\text{B}}$, where X_{A} and X_{B} are the molar fractions of A and B, respectively. Therefore, a small V^{E} requires that the pressure derivative of the interaction parameter to be small but does not necessarily means that the value of W is small. Indeed, both in this study and many other studies^{13,23}, W is assumed to be pressure independent, which is in line with the assumption that V^{E} is small.”

2) How are the uncertainties calculated and why are they not propagated into the MO calculations. This propagation needs to be performed so the model curves in figure 2 can be correctly evaluated.

Response: We actually have taken into consideration the uncertainties on molar volume difference between $\text{FeO}_{1.5}$ and FeO (ΔV) (see Table S1 for the uncertainties of the equation of state parameters), free energy of reaction ($\Delta G_r^0(P_0, T)$) (see section, **Gibbs free energy change of the reaction at 1 bar** for the uncertainties), and interaction parameters (see Table S3 for the uncertainties) and have propagated them into the calculation for oxygen fugacity. We also have mentioned in the caption of Figure 2 the 1 sigma standard deviation for oxygen fugacity (~0.5 log unit) in the original manuscript.

We have now reanalyzed the uncertainties of ($\Delta G_r^0(P_0, T)$) by performing a weighted least-squares fitting using the python package lmfit [Newville *et al.*, 2014]. In addition, we also simulate two more points for silicate melts with 12.5% iron and refit the equation of states parameters using the lmfit package (Table S1). For interaction parameters, we now include three more experimental points from Zhang *et al.*, 2019 and refit the parameters (Table S3) (see the reply to comment #14 for details).

We now explicitly mention in the text as well that we have propagated the uncertainties of all the parameters in equation (3) to the calculated oxygen fugacity and ferric iron content (Line 164). We also give explicit uncertainty range for ΔV , oxygen fugacity, and ferric iron content in the captions of Figures 1,2,3.

3) What is implied for the structure of the iron components in the silicate melt? The simulations should provide important information on melt structures which would provide at least some information upon which to evaluate the results. Why is no information on melt structure provided?

Response: We have now added a section (**Supplementary Note 1: Coordination environment of Fe in silicate melts**) and a new figure (Supplementary Figure 2 in the updated manuscript) to discuss the Fe-O coordination environment.

4) If the simulations capture correctly the thermodynamic state of the iron components in the melt why do the authors not account for the spin transitions? Why ignore this important change in electronic properties and just write it off as being small and inconsequential?- for which actually no basis is given.

The reference at line 308 is incorrect.

It is not clear what delta V is being referred to at line 312. The delta V of the HS-LS transition could well be of the order of 1-2 % if this is what the authors mean- but the effect on the delta V of equation (1) would be larger because the transitions do not occur at the same conditions. If the authors want to calculate the effects on the volume components in the lower mantle realistically, then this should be done comprehensively otherwise what is the point of doing such calculations. The argument that the MO only extends to 60 GPa is based on Ni-Co partitioning between mantle and core and is by far not accepted as a firm constraint on the depth of a MO. Figure S3 is totally misleading because it intentionally ignores the potential effects of spin transitions.

Response: Sorry for the confusion. The reference in line 308 of the previous manuscript is mislabeled. It should be Karki et al., 2018, GRL.

The reviewer is correct that the effects of spin transition may be larger on ΔV because the spin transition for Fe^{2+} and Fe^{3+} does not occur at the same conditions. We have now calculated ΔV considering the spin transition for silicate melts with 12.5% and 25% Fe at 3000 K and 4000 K. We confirm that for the most of pressures (< 100 GPa) investigated here, the uncertainties of oxygen fugacities caused by neglecting spin transition are likely to be smaller than 0.5 log unit, within the uncertainties of our model prediction.

We have now added more text in section “**Effects of spin transition of iron on ΔV and oxygen fugacity**” and move it to Method to discuss the effects of spin transition.

Both ferric and ferrous irons undergo an electronic spin transition at high pressure as predicted by a recent FPMD study⁵⁶. The high- to low-spin transition of Fe^{3+} and Fe^{2+} occurs gradually over pressure intervals centered around 90 and 110 GPa, respectively, at 3000 K. These transition pressures are higher than the maximum pressures of the magma oceans considered in this study (Fig. 2). As both Fe^{3+} and Fe^{2+} will be mostly in high spin (HS) state at relevant magma ocean pressures, we evaluate the volume difference between $\text{FeO}_{1.5}$ and FeO as $\Delta V = V_{HS}^{FeO_{1.5}} - V_{HS}^{FeO}$. However, all Fe^{3+} and Fe^{2+} will not undergo the HS-LS transition at the same condition. This means that the spin transition-induced changes in volume also contribute to our ΔV calculations. We assess the spin effects on ΔV using the spin phase diagrams from Karki et al. (2018). Considering exact HS and LS distributions for both ferrous and ferric irons, we can evaluate the volume difference between $\text{FeO}_{1.5}$ and FeO as

$$\Delta V_{\text{exact}} = (V_{HS}^{FeO_{1.5}} - V_{HS}^{FeO}) - n_{LS}^{Fe^{3+}}(V_{HS}^{FeO_{1.5}} - V_{LS}^{FeO_{1.5}}) + n_{LS}^{Fe^{2+}}(V_{HS}^{FeO} - V_{LS}^{FeO})$$

where $V_{HS}^{FeO_{1.5}} - V_{HS}^{FeO} = \Delta V$ has been rigorously constrained in this study. $n_{LS}^{Fe^{3+}}$ and $n_{LS}^{Fe^{2+}}$ represent the fractions of Fe^{3+} and Fe^{2+} in low-spin (LS) state, respectively (satisfying the relations $n_{HS}^{Fe^{3+}} + n_{LS}^{Fe^{3+}} = n_{HS}^{Fe^{2+}} + n_{LS}^{Fe^{2+}} = 1$, with $n_{HS}^{Fe^{3+}}$ and $n_{HS}^{Fe^{2+}}$ representing the corresponding HS fractions) and their values as a function of pressure and temperature for silicate melt with 25% Fe can be found in ref.⁵⁶ Karki et al. (2018) also evaluated the $V_{HS}^{FeO_{1.5}} - V_{LS}^{FeO_{1.5}}$ and $V_{HS}^{FeO} - V_{LS}^{FeO}$ to be constant with respect to pressure within their computational uncertainties. At 3000 and 4000 K, $V_{HS}^{FeO_{1.5}} - V_{LS}^{FeO_{1.5}} \approx 1.25 \text{ cm}^3/\text{mol}$ and $1.00 \text{ cm}^3/\text{mol}$, respectively, and $V_{HS}^{FeO} - V_{LS}^{FeO} \approx 1.75 \text{ cm}^3/\text{mol}$ and $1.10 \text{ cm}^3/\text{mol}$, respectively⁵⁶. We

calculate the difference of ΔV_{exact} and ΔV at 3000 and 4000 K as well as the difference of the oxygen fugacity using these two volume differences (Fig. 5).

The results show that at pressures less than 60 GPa, the deviation of volume difference caused by neglecting the spin transition is less than 3% (Fig. 5). Consequently, the difference of oxygen fugacity considering spin transition of Fe ($\log f_{\text{O}_2, \text{exact}}$) and that neglecting the spin transition ($\log f_{\text{O}_2}$) is negligible. With increasing pressure, the magnitude of $(\Delta V_{\text{exact}} - \Delta V)$ further increases and bounces back at around 100 GPa, at which the fraction of low-spin Fe^{3+} reaches around 50%. Note that at pressures greater than 80 GPa, the temperature of the MO is around 3500 K for a cold thermal profile and continues increasing with pressure. Therefore, the results at 4000 K are more relevant at these pressures. Overall, neglecting the spin transition of Fe tends to overestimate the oxygen fugacity, especially at high pressures. The maximum deviation occurs at ~ 120 GPa and is ~ 0.6 log units, around the same magnitude as the uncertainties of our model prediction (~ 0.5). Therefore, we consider the effects of spin transition of iron on the redox state of MOs to be mostly insignificant.

Figure 5. Effect of spin transition of iron of silicate melts containing 25 mol% iron at 3000, and 4000 K. **(a)** The difference of ΔV_{exact} (the molar volume difference of $\text{FeO}_{1.5}$ and FeO considering spin transition of Fe) and ΔV (the molar volume difference of $\text{FeO}_{1.5}$ and FeO considering spin transition of Fe). **(b)** The difference of oxygen fugacity considering spin transition of Fe ($\log f_{\text{O}_2, \text{exact}}$) and that neglecting the spin transition ($\log f_{\text{O}_2}$) as a function of pressure. Results are plotted along isotherms only to pressures where the simulated systems were in a liquid state.

6) The authors use their calculations simply to calculate the volumes of the iron components and then use these data in combination with one bar thermodynamic data to determine the oxygen fugacity of a melt as a function of the ferric ferrous ratio at high pressure. But why use the one bar data at all- why don't they calculate the oxygen fugacity at high pressure from their ab initio calculations directly from information on the free energy of the melt- if these calculations truly capture the thermodynamics of silicate liquids. Why use 1 bar interaction parameters, for example, when information to constrain these parameters at high pressure could be obtained through the excess volumes of mixing. If the authors cannot do this then they should state it and estimate what the uncertainty is likely to be as a result- then propagate this uncertainty in their calculation.

Response: Yes, it would be ideal to calculate the energetics of melt from first principles as well. Unfortunately, those calculations are prohibitively expensive and may also be sensitive to atomic configuration of the melts, choice of the reference system, and so on. Therefore, the energy-related terms in equation (3) are very difficult to constrain using the first-principles molecular dynamics simulations (FPMD). On the other hand, the volume of the melt can be rather precisely calculated using the FPMD method. Therefore, the strategy in our manuscript is to use experimental results to constrain the free energy term and use FPMD to constrain the volume-related term.

As far as we know, previous experimental studies have adopted a similar strategy. Direct measurements of partial molar volume of iron oxidizes in silicate melts at high pressures are prohibitively difficult. Therefore, it is a common practice to use thermodynamic model fittings rather than direct experimental measurements to constrain the partial molar volume.

We also agree that the interaction parameters and excess volume are inherently related. One could, in principle, calculate the interaction parameters from the excess volume by integrating over $V^E = \left(\frac{\partial G_{\text{mix}}}{\partial P}\right)_T = \left(\frac{\partial H_{\text{mix}}}{\partial P}\right)_T$ if one anchor point of the excess free energy is given. However, we do not determine the V^E in this study. Instead, we determine ΔV , which is different from V^E (see the reply to comment #1 for further details). Also, the anchor point G_{mix} value is also unknown and hard to determined using FPMD as discussed above.

We have also considered the uncertainties in all parameters in equation 3, including the uncertainties on molar volume difference between $\text{FeO}_{1.5}$ and FeO (ΔV), free energy of reaction ($\Delta G_r^0(P_0, T)$), and interaction parameters and have propagated them into the calculation of oxygen fugacity.

7) How can the authors be sure that their determined volume change describes and encapsulates completely the expected changes in the ferric ferrous components in the liquid as a function of pressure? For experimental data the fitting of a volume change to parameterize the pressure behavior of the ferric/ferrous ratios is reasonable because the data themselves actually constrain the variations in the ratio whereas the volume is just a fitting term and may not actually correspond to the standard state volume change. There could be other factors that end up being inadvertently described by the volume term such as changes in the interaction parameters with pressure. For the experiment based models this does not matter because they are fitted to the experiments and reproduce those values but in this study the authors attribute all changes with pressure to their estimated volume changes. The extent to which they can be sure that the calculated volumes encapsulate the real ferric/ferrous ratio variations needs to be discussed in detail in the SI. Can the authors be sure that changes in electronic state, entropies and excess volumes with pressure are being meaningfully accounted for just through the volume change and if not what are the uncertainties?

Response: We now have added a section (**Calculation of volume difference**) in the Method on how we determine ΔV . The partial molar volumes have strict thermodynamic meaning. We believe that the way of calculating ΔV is thermodynamically correct.

Under the *NTV*-canonical ensemble, at a given volume of the system, the pressure and other thermodynamic terms are calculated by solving the many-body Schrodinger equation. During this process, the changes in electronic state and other thermodynamic have been meaningfully considered. The output pressure term contains uncertainties which are propagated into the equation of state parameters (see Supplementary Table 1).

8) The authors propose depths of “the” magma oceans for which they calculate fo₂ gradients. For the

earth this is taken as 55 GPa using single stage models for metal silicate equilibration. The use of this value so unequivocally in the manuscript is naïve as there are many alternative hypotheses as to how the Earth obtained its Ni-Co ratio on which this estimate is based. This implies also, for example, that there was a single magma ocean with a particular depth at which metal equilibrated with the “entire” magma ocean and then no further equilibration with metal occurred in lower pressure magma oceans. Recent models that have examined this, such as Deguen et al., 2014, EPSL 391: 274-287, show quite clearly that it is unrealistic to assume whole mantle equilibration with metal in a magma ocean. Only part of the magma ocean would reequilibrate with iron metal as it descends. A simplified scenario is probably ok as long as the simplifications and the assumptions are all clear but to maintain dogmatically that “the” MO on earth was at 55 GPa is simply counter to a lot of evidence. There is too little discussion at the moment over the actual scenario that is being implied. Assumptions such as the “entire” magma ocean equilibrating with iron at 55 GPa as the last step in significant core formation are not explicitly stated in the manuscript and need to be explained.

Response: We use 55 GPa as an example to illustrate the oxygen fugacity profile of the magma ocean (Figure 2). But we did consider a wide pressure range for the bottom of the magma oceans (Lines 239 – 250 in the old manuscript). We agree that the pressure value of 55 GPa is based on single stage model and assumes complete equilibrium between the silicate melt and iron melt. Indeed, in the previous manuscript, we mention that the “In contrast, the Earth’s MO may reach 25-90 GPa based on moderately siderophile elements abundances, assuming models for single or multi-stage core formation with partial or complete equilibrium between impactor and proto Earth. (Lines 244-246 in the old manuscript)” and we did consider the variation of the magma ocean depth on the redox state of the magma ocean. Actually, the whole Figure 3 and Figure S7 in the previous manuscript (Figure 3 and Supplementary Figure 10 in the updated text) are for this purpose.

We have now added more text to explicitly state that the 55 GPa is based on single stage model for metal silicate equilibration in Lines 178-181 of the updated manuscript, as follows:

“The pressures considered here are based on single stage model and the complete equilibrium between the silicate melt and iron melt. More general consideration of magma ocean depths is discussed below and in Fig. 3. ”

9) This raises the further question of what is the actual scenario that the authors are proposing. How is the ferric iron content of the magma ocean actually increased between the moon, mars and Earth? How does oxygen mass balance in their calculation? Presumably the authors are implying that before the magma ocean formed the ferric iron content of the mantle was very low- although they do not specifically state this. Therefore where does the oxygen come from to create the raised ferric iron levels in the magma ocean? Buffering with iron metal has the implication that material either initially more oxidised or more reduced attains the same level of oxidation i.e. ferric iron content after equilibration with iron metal. If material that formed the earth was initially more oxidised then it is clear that reaction with iron metal would lower the ferric iron content to a specific level by reacting $\text{Fe} + \text{Fe}_2\text{O}_3$ into FeO . But how does this buffering work if the material is more reduced before it interacts with iron metal. How does the redox buffering actually operate? This is not at all clear in the manuscript.

Response: We followed Zhang et al., 2017, GCA and Hirschmann 2012, EPSL (see his Figure 3 for a cartoon) and considered that at the bottom magma ocean, the silicate equilibrates with the iron pond; at shallower depths, the magma ocean is chemically homogenized by vigorous convection. We did not consider how the magma ocean reached this state and did not consider what the redox of the magma ocean was before this state. The scenario we consider is that the magma ocean and the iron pond have already reached the thermodynamic equilibrium. In this case, the redox in the magma ocean is buffered by the reaction of Fe^{2+} and O_2 to form Fe^{3+} (equation 1 in the main text).

How the magma ocean reaches this state and whether magma ocean was more oxidized or more reduced are interesting questions and await further investigation. However, we argue that these are not important for the study because once the new thermodynamic equilibrium state is established (the state depicted as Figure 3 in Hirschmann 2012, EPSL), the redox state of whole magma ocean is reset. This reset process could be achieved by a giant impact or by the accretion of planetesimals followed by vigorous thermal and chemical exchanges. The proto-earth is an open system and addition of exotic materials may also modify its thermodynamic state.

To conclude, how the magma ocean reaches the equilibrium with the iron pond in the bottom is interesting itself. But here we focus on the time snapshot when the equilibrium is reached.

10) Armstrong et al created a model that fitted their data over the pressure range of their experiments. They did not then extrapolate this model and they state quite clearly that the properties likely change at higher pressure. It is simply not scholarly to take a model that was only calibrated and derived over a particular pressure regime and extrapolate it to over double the pressure and a much higher temperature as in Figure 2. This model and certainly that of O'Neill was never intended to be employed in that way and by doing so the authors intentionally make the models look erroneous. Furthermore the uncertainties of all these extrapolations would make them meaningless so it is questionable what purpose this serves. These models were based on experimental measurements of ferric iron contents and oxygen fugacities for which there is almost certainly a range of equation of state parameters that would fit the data- but that would extrapolate very differently. The main point of the experimental based models is that they allow the data to be interpolated as a function of oxygen fugacity over the pressure range at which they were calibrated.

Line 225 “mainly due to their unwarranted extrapolations from low P/T experimental conditions”. This statement is totally misleading in its use of the word “their” and could be understood in two ways. What the authors hopefully mean is that they themselves are performing an unwarranted extrapolation- which they should not do and the curve should simply be removed. One could understand from this statement that Armstrong et al themselves extrapolated the data- which is not the case. The statement is misleading and should be clarified and the authors need only state that no meaningful results can be obtained by extrapolating the model so far out of the range at which it was determined.

Response: We agree that the extrapolating the experimentally determined models out of the experimental conditions are dangerous. We apologize for the confusion we made in the sentence in Line 225 in the previous manuscript.

We have now only used the previous models within their applicable pressure ranges. We have revised the text and Figure 2 accordingly.

11) Line 164 ref 32 is used to calculate IW however again the authors are extrapolating this equation of state to much higher temperatures than it was intended. This should be addressed.

Response: Thank you for pointing this out. We have now stated explicitly in the main text that the temperatures considered are higher than this IW buffer is calibrated. We extrapolate this buffer equation at high temperatures following the conventions (Zhang et al., 2017; Armstrong et al., 2019), which would facilitate the comparison between this study and previous ones (Lines 170-172).

12) Line 166 the authors use previous values for the fo₂ relative to IW for the earth but they should be able to calculate these values using their model and state clearly what the assumptions are. These fo₂s can then be made internally consistent with the compositions in table S4.

Response: This study established a model for the redox buffering between Fe^{2+} and Fe^{3+} . However, to calculate the redox state of the bottom of the magma ocean, one needs the model for redox buffering between FeO in the silicate melt and Fe melt in the iron pond, which is beyond the scope of this study. We therefore take the canonical values of redox states at the bottom of the magma oceans as the anchor point to resolve the redox state of the magma oceans.

13) Figure S5 is not useful as it provides no information on the conditions at which the fit is good or bad other than the information “1 bar” and “high pressure”. The use of the $\text{Fe}^{3+}/\text{Fe}^{2+}$ ratio also makes it impossible to usefully evaluate the accuracy of the model. In these figures all the experimental data at high pressures seem to fall on a slope that implies some systematic deviation to the model- but it is impossible to evaluate this without a comparison that provides also P, T and f_{O_2} information which must be facilitated. Rather than showing 4 fits- that all seem to be identical anyway- the authors should allow the model to be evaluated against the actual high pressure experimental measurements of Fe^{3+} over total iron. As most of the experimental data is determined at a buffered oxygen fugacity- the model should be compared against the experimental data at the particular f_{O_2} . Showing $\text{Fe}^{3+}/\text{Fe}^{2+}$ ratio is immediately difficult to assess because it varies over 3 orders of magnitude particularly due to the changes in the total iron content of the 1 bar experiments. The authors should rather evaluate the $(\text{Fe}^{3+}/\text{Total iron})$ ratio differences between the data and model either by comparing the model values with the data points themselves or by plotting the miss fit as a function of P with information on T and f_{O_2} given.

Response: We followed the convention of Figure 10 of Jayasuriya et al., 2004 to plot the $\text{Fe}^{3+}/\text{Fe}^{2+}$ ratio. This figure shows the goodness of fitting of all four different models we choose. Following the reviewer’s suggestion, we have now added a section in supplementary information (**Model prediction vs. high-pressure experiments**) to compare the model prediction against the high-pressure experimental results. We also added a figure to show the ferric iron content not only as a function of pressure but also temperature. From this analysis, we confirm that our predicted values are broadly consistent with extant experimental results.

14) Line 499 “we use the value after correction for crystallization”- it is not clear what this means or why the authors choose to exclude some experimental data. This has to be more clearly explained otherwise it looks like they may be excluding experimental data that do not agree with their model.

Response: Thank you for pointing this out.

In the original manuscript, the experimental data we used were S6928, S6889, S6820, Z1468, Z1621, Z1666, S6654, S6776. We used the $\text{Fe}^{3+}/\Sigma\text{Fe}$ ratios reported in Table 4.4 of Armstrong, 2018 (PhD thesis) and in Table S3 of Armstrong et al., 2019. The choice of the data followed Armstrong, 2018 (PhD thesis) and Armstrong et al. 2019. Samples S6879, S6811, S6510, and S6977 are excluded because they suffered from inhomogeneous Pt contamination of the oxygen buffer and are classified as unreliable results by Armstrong, 2018 (PhD thesis) and Armstrong et al. 2019. We also excluded the sample S6606 because the temperature was estimated, not measured. We excluded sample Z1794 following the Figure 4.6 of Armstrong, 2018 (PhD thesis). The reason is probably due to that for this sample, Ru was added only as metal rather than metal and oxidized as other experimental runs. However, we do notice that in the Figure 1 of Armstrong et al. 2019, this sample is taken into account. We also excluded samples, Z1971, and Z1850 because these two samples do not reach superliquidus. We do note that we accidentally omitted sample S6973.

Now we have now taken into account all the data shown in Figure 1 of Armstrong et al., 2019. They are S6928, S6889, S6820, Z1794, Z1468, Z1621, Z1666, S6654, S6606, S6776, and S6973 (in total 11

samples). And we consider both the data reported in the Figure 1 of Armstrong et al, 2019 and also the corresponding ones with possible Ru contribution removed shown in Figure S7 of Armstrong et al, 2019.

With these new datasets, we have re-fitted the interaction parameters and found that the results barely change (Supplementary Figure 6 and Supplementary Table 3 in the updated manuscript).

We have now discussed the how the experimental data are selected in the **Model prediction vs. high-pressure experiments** section in the supplementary information as follows,

“The previous high-pressure experimental data are selected as follows. We adopt all the six high pressure data by ref. ²³. We also use all 19 experimental results by ¹³. For the recent high pressure study by ref. ¹², we take into account all the data shown in Figure 1 of Armstrong et al., 2019 and the corresponding ones with possible Ru contribution removed shown in their Figure S7. They are S6928, S6889, S6820, Z1794, Z1468, Z1621, Z1666, S6654, S6776, Z1794, and S6973 (in total 11 samples). Samples S6879, S6811, S6510, and S6977 are excluded because they suffered from inhomogeneous Pt contamination of the oxygen buffer and are classified as unreliable results by ref. ¹².”

Reviewers' Comments:

Reviewer #2:

Remarks to the Author:

I appreciate the time and effort Deng et al. have taken to revise their manuscript "A magma ocean origin to divergent redox evolutions of Earth, Mars, and the Moon: Implications for early atmospheres". I had an overall positive response to the initial submission, and I believe the authors have addressed the issues I raised. I would like to reiterate some criticisms raised by the other two reviewers.

One reviewer correctly stated that "the essential premise and conclusions of the paper are not new". I agree, and would characterize this work as substantiating, extending, and clarifying previous work. I therefore believe the contributions are sufficient to warrant publication.

Similarly, another reviewer raised many technical questions about the computational methodology in the paper. In response, the authors have done some additional work, but for the most part just provided more detailed explanations of what they had already done. Therefore, I would defer to that reviewer's judgement as to whether the revised manuscript provides enough detail so as to be fully scrutinizable and replicable.

Reviewer #3:

Remarks to the Author:

I find the paper to be improved with far better descriptions of the method and the unwarranted extrapolation of other experimental based models has now been removed. I see no major objection to publishing it.

I still think their assignment of all thermodynamic pressure dependence simply to the volume change is potentially incorrect but they never recognize this and are still rather dismissive (Page 9 answer to comment 7). It is fine if they want to do this but I think it is important to point out how easily this could be incorrect. Attributing the change in Gibbs free energy with pressure to the integral of the volume for an end member pure mineral phase is fine because only elastic changes take place as the system moves from one state of pressure to another. However imagine now we have a mineral phase with multiple sites of mixing where the occupation of those sites changes with pressure. There is no way that the volume change of the mineral phase with pressure now accounts for the entire change in free energy because at the very least there are configurational entropy changes not to mention excess enthalpies. Now imagine that instead of a multi-site mineral phase you have a melt - it is not such a stretch of the imagination to see that the volume cannot account for the entire contribution to the free energy. Put very simply- different configurations with different inherent entropies can have the same volume. But would have different free energies. Ok second even simpler example. Think of a mineral phase going through multiple 1st order phase transitions with pressure such that a series of structures with different atom coordination occurs. Can that series of transitions be purely described through the volume changes? But a melt does exactly the same thing.

Experimental studies attribute pressure changes simply to the end member volume change because they are using the volume change as a pressure fitting parameter. But that does not mean that all the free energy change is really encapsulated by the end member volumes. However experiments also have experimental observations of the actual f_{O_2} - Fe^{3+} relation at high pressure- something which the current study does not have. I add this as a warning that treating the oxygen fugacity pressure variation in this simplistic manner is probably not correct- experiments do not do this- they are fitting and interpolating between actual f_{O_2} and Fe^{3+} measurements- which is different! The authors break the problem down to be a question of volume changes only- but it is not - it is a question of how the Fe^{3+} /total Fe content of a melt changes with pressure and oxygen fugacity.

Lines 264-266 need to be rewritten as the grammar is poor.

Supplementary Figure 6 is still waste of time because the scale is too small to examine the high pressure data agreement which is basically the only part that is important if you have measured the volume change. There is a large about of data at low $3+/2+$ ratio that also does not agree with the model but there is no discussion made to explain why this may be.

Adding S Figure 8 is a great improvement- but then that also opens the question why the model does not match at all the data of O'Neill et al., 2006. This is one of the most carefully performed high pressure studies and misses the current model completely. The authors suggest there may be precision problems in Mössbauer spectra- but x-ray spectroscopic techniques on the other hand are extremely sensitive to calibrations which are ultimately calibrated against- yep - Mössbauer data

Reviewers' comments:

Reviewer #2 (Remarks to the Author):

I appreciate the time and effort Deng et al. have taken to revise their manuscript "A magma ocean origin to divergent redox evolutions of Earth, Mars, and the Moon: Implications for early atmospheres". I had an overall positive response to the initial submission, and I believe the authors have addressed the issues I raised. I would like to reiterate some criticisms raised by the other two reviewers.

One reviewer correctly stated that "the essential premise and conclusions of the paper are not new". I agree, and would characterize this work as substantiating, extending, and clarifying previous work. I therefore believe the contributions are sufficient to warrant publication.

Response: Thank you for recognizing our contributions.

Similarly, another reviewer raised many technical questions about the computational methodology in the paper. In response, the authors have done some additional work, but for the most part just provided more detailed explanations of what they had already done. Therefore, I would defer to that reviewer's judgement as to whether the revised manuscript provides enough detail so as to be fully scrutinizable and replicable.

Response: Thank you for noting our effort on elaborating the method used.

Reviewer #3 (Remarks to the Author):

I find the paper to be improved with far better descriptions of the method and the unwarranted extrapolation of other experimental based models has now been removed. I see no major objection to publishing it.

Response: Thank you for considering our revision.

I still think their assignment of all thermodynamic pressure dependence simply to the volume change is potentially incorrect but they never recognize this and are still rather dismissive (Page 9 answer to comment 7). It is fine if they want to do this but I think it is important to point out how easily this could be incorrect. Attributing the change in Gibbs free energy with pressure to the integral of the volume for an end member pure mineral phase is fine because only elastic changes take place as the system moves from one state of pressure to another. However imagine now we have a mineral phase with multiple sites of mixing where the occupation of those sites changes with pressure. There is no way that the volume change of the mineral phase with pressure now accounts for the entire change in free energy because at the very least there are configurational entropy changes not to mention excess enthalpies. Now imagine that instead of a multi-site mineral phase you have a melt - it is not such a stretch of the imagination to see that the volume cannot account for the entire contribution to the free energy. Put very simply- different configurations with different inherent entropies can have the same volume. But would have different free energies. Ok second even simpler example. Think of a mineral phase going through multiple 1st order phase transitions with pressure such that a series of structures with different atom coordination occurs. Can that series of transitions be purely described through the volume changes? But a melt does exactly the same thing. Experimental studies attribute pressure changes simply to the end member volume change because they are using the volume change as a pressure fitting parameter. But that does not mean that all the free energy change is really encapsulated by the end member volumes. However experiments also have

experimental observations of the actual f_{O_2} –Fe³⁺ relation at high pressure- something which the current study does not have. I add this as a warning that treating the oxygen fugacity pressure variation in this simplistic manner is probably not correct- experiments do not do this- they are fitting and interpolating between actual f_{O_2} and Fe³⁺ measurements- which is different! The authors break the problem down to be a question of volume changes only- but it is not – it is a question of how the Fe³⁺/total Fe content of a melt changes with pressure and oxygen fugacity.

Response: Thanks for pointing this out.

To clarify this, we add the following sentences in the main text (Lines 152-156):

“The above equation has been widely used in many literatures^{3,12,13,23} and it suggests that the variation of f_{O_2} with pressure explicitly hinges on ΔV only. However, one should note that $\Delta V(P, T)$ not only depends on pressure and temperature but also implicitly on many extensive properties, including configuration entropy and excess enthalpy.”

We agree with what reviews #3 has to say, which, we believe, is not mutually exclusive with our way of dealing with the pressure dependency of oxygen fugacity. Within our framework, we can attribute the pressure dependency of the oxygen fugacity to ΔV only. And ΔV implicitly includes the effects of pressures on configuration entropy and excess enthalpy.

Our approach has already considered pressure dependency of oxygen fugacity on extensive properties including entropy and enthalpy. Please note that we have evaluated $\Delta V(P, T)$ directly from first-principles MD simulations of FeO- and FeO_{1.5}-bearing silicate melts over wide volume ranges at different temperatures and for different bulk compositions.

We now explain our approach in detail as follows. We start by rewriting the thermodynamic relationship for the buffer reaction of interest

as,

$$-\frac{\Delta G_r^0(P_0, T) + \int_{P_0}^P \Delta V(P, T) dP}{RT} = \ln \frac{X_{\text{FeO}_{1.5}}^{\text{melt}}}{X_{\text{FeO}}^{\text{melt}}} + \ln \frac{\gamma_{\text{FeO}_{1.5}}^{\text{melt}}}{\gamma_{\text{FeO}}^{\text{melt}}} - \frac{1}{4} \ln f_{O_2} \quad (2)$$

where $\Delta G_r^0(P_0, T)$ is the free energy of the reaction at reference pressure P_0 (1 bar) and temperature T . ΔV is the molar volume difference between FeO_{1.5} and FeO in the melts. X and γ are the molar fractions and activity coefficients of the Fe-oxide component, respectively, f_{O_2} is the oxygen fugacity, and R is the gas constant.

The equation (2) is thermodynamically rigorous and has been widely used in numerous literatures (e.g., Armstrong et al., 2019; Hirschmann, 2012; O’Neill et al., 2006). Let us derive the equations (2) in order to explain why one can assign the pressure dependence of f_{O_2} to ΔV . The derivation can also be found in some thermodynamic textbooks (Chapter 11 of Dehoff, 2006, Thermodynamics in Materials Science; Chapters 2 and 10 of Karato, 2008, Deformation of Earth Materials). At thermodynamic equilibrium, the chemical potentials of left- and right-hand side terms should be equal,

$$\mu_{\text{FeO}}^{\text{melt}}(P, T) + \frac{1}{4} \mu_{O_2}(P, T) = \mu_{\text{FeO}_{1.5}}^{\text{melt}}(P, T) \quad (3)$$

The chemical potential at pressure P can be related to that at pressure P_0 by

$$\mu_{\text{FeO}}^{\text{melt}}(P, T) = \mu_{\text{FeO}}^{\text{melt}}(P_0, T) + \int_{P_0}^P v_{\text{FeO}}^{\text{melt}}(p, T) dp \quad (4a)$$

$$\mu_{\text{FeO}_{1.5}}^{\text{melt}}(P, T) = \mu_{\text{FeO}_{1.5}}^{\text{melt}}(P_0, T) + \int_{P_0}^P v_{\text{FeO}_{1.5}}^{\text{melt}}(p, T) dp \quad (4b)$$

$$\mu_{O_2}(P, T) = \mu_{O_2}(P_0, T) + RT \ln f_{O_2} \quad (4c)$$

Furthermore, the chemical potential of a component as a function of concentration X can be related to its pure endmembers by,

$$\mu_{FeO_{1.5}}^{melt}(P_0, T) = \mu_{FeO_{1.5}}(P_0, T) + RT \ln \gamma_{FeO_{1.5}}^{melt} X_{FeO_{1.5}}^{melt} \quad (5a)$$

$$\mu_{FeO}^{melt}(P_0, T) = \mu_{FeO}(P_0, T) + RT \ln \gamma_{FeO}^{melt} X_{FeO}^{melt} \quad (5b)$$

where $\mu_{FeO_{1.5}}$ and μ_{FeO} are the chemical potential of the pure endmembers, i.e., $FeO_{1.5}$ melt and FeO melt, respectively. Insert equation (4a-c) and (5a-b) in equation (3) to get

$$\begin{aligned} & \mu_{FeO}(P_0, T) + RT \ln \gamma_{FeO}^{melt} X_{FeO}^{melt} + \int_{P_0}^P v_{FeO}^{melt}(p, T) dp + \frac{1}{4} \mu_{O_2}(P_0, T) + \frac{1}{4} RT \ln f_{O_2} \\ & = \mu_{FeO_{1.5}}(P_0, T) + RT \ln \gamma_{FeO_{1.5}}^{melt} X_{FeO_{1.5}}^{melt} + \int_{P_0}^P v_{FeO_{1.5}}^{melt}(p, T) dp \quad (6) \end{aligned}$$

Re-arrangement of the above equation yields the equation (2) with

$$\Delta G_r^0(P_0, T) = \mu_{FeO_{1.5}}(P_0, T) - \mu_{FeO}(P_0, T) - \frac{1}{4} \mu_{O_2}(P_0, T) \quad (7a)$$

$$\Delta V(P, T) = v_{FeO_{1.5}}^{melt}(P, T) - v_{FeO}^{melt}(P, T) \quad (7b)$$

Using the equation (2), one can further derive the pressure dependency of the oxygen fugacity as

$$\left(\frac{\partial \ln f_{O_2}}{\partial P} \right)_{T, X_{FeO_{1.5}}^{melt}, X_{FeO}^{melt}} = 4 \Delta V \times \left(\frac{\partial \Delta V}{\partial P} \right)_{T, X_{FeO_{1.5}}^{melt}, X_{FeO}^{melt}} \quad (8)$$

Equations 5a-b show that the “configurational entropy” or “excess enthalpies” terms are taken into account in these terms: $RT \ln \gamma_{FeO_{1.5}}^{melt} X_{FeO_{1.5}}^{melt}$ and $RT \ln \gamma_{FeO}^{melt} X_{FeO}^{melt}$. Here, we assume these terms do not have explicit

pressure dependency, that is $\left(\frac{\partial RT \ln \gamma_{FeO_{1.5}}^{melt} X_{FeO_{1.5}}^{melt}}{\partial P} \right)_{T, X_{FeO_{1.5}}^{melt}, X_{FeO}^{melt}} = 0$. Yet, “configurational entropy” and

“excess enthalpies” should vary with pressure, as reviewer #3 suggested. Instead, the pressure effects on “configurational entropy” and “excess enthalpies” all go into $v_{FeO}^{melt}(P, T)$ and $v_{FeO_{1.5}}^{melt}(P, T)$. This can be understood as follows: μ_{FeO}^{melt} is a state function and thus the specific value of μ_{FeO}^{melt} does not depend on the path we take from state (P_0, T) to the target state (P, T) . Here we take a path where we “mix” FeO in the melt with other components at reference pressure P_0 and target temperature T and then “increase” the pressure to the target P but keeping all other parameters intact. Of course, one can take other paths. For example, one can first increase pressure from P_0 to P and then “mix” FeO with other components. Alternatively, one can increase pressure and “mix” FeO with other components simultaneously as the reviewer #3 suggested. No matter which path one takes, the final μ_{FeO}^{melt} must be the same since it is a state property.

More specifically, the variation of “configurational entropy” and “excess enthalpies” can cause the variation of partial volume of FeO . Then one can express v_{FeO}^{melt} as function of configurational entropy and excess enthalpy. Moreover, extensive properties like “configurational entropy” and “excess enthalpies” are ultimately controlled by the variation of intensive properties such as pressure and temperature for a given system, we can then write v_{FeO}^{melt} as a function of pressure and temperature only, i.e., $v_{FeO}^{melt}(P, T)$. Mathematically, the above discussion can be expressed by rewriting the equation (8) as

$$\begin{aligned} \left(\frac{\partial \ln f_{O_2}}{\partial P} \right)_{T, X_{FeO_{1.5}}^{melt}, X_{FeO}^{melt}} &= 4\Delta V \times \left(\frac{\partial \Delta V}{\partial P} \right)_{T, X_{FeO_{1.5}}^{melt}, X_{FeO}^{melt}} \\ &= 4\Delta V \times \left(\frac{\partial \Delta V}{\partial H_{ex}} \frac{\partial H_{ex}}{\partial P} + \frac{\partial \Delta V}{\partial S} \frac{\partial S}{\partial P} + \dots \right)_{T, X_{FeO_{1.5}}^{melt}, X_{FeO}^{melt}} \end{aligned}$$

, where S and H_{ex} are the “configurational entropy” and “excess enthalpy” respectively; ... represents the product of the partial derivative of ΔV with respect to any other extensive properties and the derivative of extensive properties to pressure.

In conclusion, we choose to calculate chemical potential μ without explicitly considering the pressure dependency of properties such as configuration entropy and excess enthalpies. Within this framework, we can attribute the pressure dependency of the oxygen fugacity as a function of ΔV only. Thus, ΔV implicitly includes the effects of pressures on configuration entropy and excess enthalpy because the variation of these extensive properties also affects the ΔV . The different ΔV at same P , T for $MgSiO_3$ melt with different iron content presented in this study is one example that the chemical environment (composition, configuration, mixing) of Fe-oxide would affect the ΔV .

Lines 264-266 need to be rewritten as the grammar is poor.

Response: We have revised the corresponding sentence as, “However, this ferric iron content profile likely evolves during the solidification of the MO. The evolution is controlled by how the MO crystallizes out and the partitioning of iron species between the melt and crystal, which are so far poorly constrained. Nevertheless, our study suggests that the whole mantle of Earth and Mars could have been as enriched in ferric iron as the present-day upper mantle since the MO stage.”

Supplementary Figure 6 is still waste of time because the scale is too small to examine the high pressure data agreement which is basically the only part that is important if you have measured the volume change.

Response: We appreciate Reviewer #3’s suggestion, but we feel that Supplementary Figure 6 does provide critical information to evaluate the activity coefficients and reference free energy term. Supplementary Figure 8 indeed show the comparison at high pressures, which is complementary to Supplementary Figure 6. To clarify this, we add the following discussion in Supplementary Note 4 (Lines 557-560).

“Note that Supplementary Fig. 6 is not intended to examine the high-pressure data, but it provides valuable comparison between the model prediction and experimental results at 1 bar. More detailed comparison between the model prediction and high-pressure experiments is considered in Supplementary Fig. 8.”

We further illustrate the above points in detail below.

We include 225 data points at 1 bar and 32 data points at higher pressure in Supplementary Figure 6. The comparison between our model prediction and the 225 1-bar experimental data helps evaluate the activity coefficients and reference free energy, i.e., $\Delta G_r^0(P_0, T)$, which are as critical as the volume change (ΔV) in order to constrain the ferric iron content. We emphasize that in this study we not only calculated the ΔV but also fitted activity coefficients and determined the reference free energy. Therefore, we prefer keeping this figure in order to evaluate the activity coefficients and reference free energy term determined in this study. Supplementary Figure 6 is not intended to examine the high-pressure data, but it provides valuable comparison between the model prediction and experimental results at 1 bar. Instead, supplementary Figure 8 provides complementary information for high pressures. In addition, some previous studies plot the model predicted ferric iron content against the experimental results in the same manner (for example, Figure 10

of Jayasuriya et al., 2004). It is easier for readers to compare our model with previous ones with this figure. Nevertheless, we leave it to the editor to decide if we need remove Supplementary Figure 6.

There is a large amount of data at low $3+/2+$ ratio that also does not agree with the model but there is no discussion made to explain why this may be.

Response: Thanks for pointing this out. We have now discussed the above points in the Supplementary Note 4 (Lines 573-585), also stated as follows:

“For low Fe^{3+}/Fe^{2+} samples, our model predictions (Fit 1,2,3,4 in Supplementary Figure 6) deviate notably from some experimental results, mostly by Kress and Carmichael (1988). This disagreement may be caused by 1) inaccurate experimental results for ferric iron content of those samples by Kress and Carmichael (1988). Indeed, Kress and Carmichael (1991) remeasured the ferric iron content of some samples in Kress and Carmichael (1988) and found the new results differ significantly from those reported by Kress and Carmichael (1988); 2) the lack of MnO component in our model. The MnO contents of those samples exhibit a moderately positive correlation with the deviations of model predictions, indicating that MnO may be a key component that controls the Fe^{3+}/Fe^{2+} ratio when the Fe^{3+} is low. Future studies are warranted to explore this effect. For this study, due to the insufficient experiments with MnO bearing samples, we are not able to include the MnO component in our model. Nevertheless, MnO is a trace/minor component for Earth, Mars, and the Moon and thus not of interest here.”

We further illustrate the above points in detail below.

As the reviewer pointed out, for low Fe^{3+}/Fe^{2+} samples, our model prediction ($R_{pred} = Fe^{3+}/Fe^{2+}$ predicted by our model) deviates significantly from the experimental results ($R_{exp} = Fe^{3+}/Fe^{2+}$ measured experimentally). There are two possible reasons.

1) Uncertainty of experiments. It is noted that for those samples, although the R_{pred} / R_{exp} deviates largely from 1, the absolute differences $|R_{pred} - R_{exp}|$ are generally very small, around or less than 0.04-0.06. These samples are mostly from Kress and Carmichael (1988). Although the specific values of experimental errors were not given in the source, their Figure 1 suggest that the uncertainties of R_{exp} is quite substantial, especially at low oxygen fugacity (thus low R_{exp}). We note that Kress and Carmichael (1991) re-measured the ferric iron content of some samples from Kress and Carmichael (1988) and found the new results differ from the previous ones by 20-30%, “clearly outside the range of analytical error”. Although Kress and Carmichael (1991) did not re-measure those low Fe^{3+}/Fe^{2+} samples of Kress and Carmichael (1988), the reported results of those samples may suffer from the same analytical errors.

2) This discrepancy may be caused by the deficiency of our model. As discussed in the Supplementary Note 4, concentrations of some oxides like Na_2O , TiO_2 , P_2O_5 , V_2O_3 , Cr_2O_3 , and MnO are generally minor and vary marginally. The small range of composition of these component results in the unresolvable or statistically insignificant Margules interaction parameters. As such, in the best model “Fit 3”, we force the interaction parameters for those components to be 0. This choice inevitably introduces error in the model prediction, especially for those with large and varying Na_2O , TiO_2 , P_2O_5 , V_2O_3 , Cr_2O_3 , and MnO contents. To further explore which component excluded in our model causes the disagreement between R_{pred} and R_{exp} for low R_{exp} samples, we plot the concentration of Na_2O , TiO_2 , P_2O_5 , V_2O_3 , Cr_2O_3 , and MnO against the deviation of model prediction ($|R_{pred} - R_{exp}| / R_{exp}$) for low R_{exp} samples ($R_{exp} \leq 10\%$) from Kress and Carmichael (1988) in Figure 1 below. The concentration of MnO shows a pronounced positive correlation with the prediction deviation. The contents of P_2O_5 , Na_2O , and TiO_2 exhibit relatively weaker negative correlations. Therefore, the lack of MnO in our model may be the major cause for the inaccuracy of our model at low R_{exp} .

To our best knowledge, many previous 1-bar models (e.g., Jayasuriya et al., 2004; Kress et al., 1991; Kress et al., 1988) fail to predict the ferric iron content of low R_{exp} sample by Kress and Carmichael (1988). None of these models takes into account the MnO component, supporting our speculation that MnO may be a key component that controls the $\text{Fe}^{3+}/\text{Fe}^{2+}$ ratio when the Fe^{3+} is low. Future studies are warranted to explore this effect.

To sum up, the disagreement between our model prediction and experimental results for $\text{Fe}^{3+}/\text{Fe}^{2+}$ ratio of low Fe^{3+} content samples may be caused by 1) inaccurate experimental results for ferric iron content for those samples by Kress and Carmichael (1988) or 2) the lack of MnO component in our model. A possible way to test the 2nd hypothesis is to conduct experiments with samples of various MnO contents at very reduced conditions and compare the results with our model prediction. If the 2nd hypothesis holds, we would expect a good correlation between the model deviation from the experimental results with the MnO concentration. For this study, due to the insufficient experiments with MnO bearing samples, we are not able to include the MnO component in our model. Nevertheless, the MnO is a trace/minor component for Earth (~ 0.12 wt%, McDonough and Sun, 1995), Mars (amount not well constrained), and the Moon (~ 0.15 wt%, Elardo et al., 2011) and thus not of interest in this study.

Figure 1. Concentration of components in silicate melts excluded in our model (Fit 3) vs. the prediction deviations ($|R_{\text{pred}} - R_{\text{exp}}|/R_{\text{exp}}$). The correlation between concentration of MnO, Na_2O , P_2O_5 , TiO_2 , V_2O_3 , and Cr_2O_3 , and the prediction deviation are 0.58, -0.37, -0.45, -0.39, -0.14, 0.08.

Adding S Figure 8 is a great improvement- but then that also opens the question why the model does not match at all the data of O'Neill et al., 2006. This is one of the most carefully performed high pressure studies and misses the current model completely. The authors suggest there may be precision problems in Mössbauer spectra- but x-ray spectroscopic techniques on the other hand are extremely sensitive to calibrations which are ultimately calibrated against- yep - Mössbauer data

Response: Sorry for confusion. Reviewer #3 raises an important point. We have revised the corresponding sentence (Lines 631-632) as: “Zhang, et al. (2017) suggested that this discrepancy may result from the precision problems when sextets present in Mössbauer spectra, which is the case for all of data by O'Neill et al. (2006)’. Yet, future studies are needed to clarify this discrepancy, both between our model prediction and the experiments by O'Neill et al. (2006) and between experiments by Zhang et al. (2017) and those by O'Neill et al. (2006).

Reviewers' Comments:

Reviewer #3:

Remarks to the Author:

The manuscript is now fine to publish

REVIEWERS' COMMENTS:

Reviewer #3 (Remarks to the Author):

The manuscript is now fine to publish

Response: Thank you for your support.